# Integrin alpha11 is an Osteolectin receptor and is required for the maintenance of adult skeletal bone mass

Bo Shen[1], Kristy Vardy[1], Payton Hughes[1], Alpaslan Tasdogan[1], Zhiyu Zhao[1], Rui Yue[2], Genevieve M Crane[1], Sean J Morrison[1,3,4]*

[1]Children's Research Institute, University of Texas Southwestern Medical Center, Dallas, United States; [2]Institute of Regenerative Medicine, Shanghai East Hospital, Shanghai Key Laboratory of Signaling and Disease Research, School of Life Sciences and Technology, Tongji University, Shanghai, China; [3]Department of Pediatrics, University of Texas Southwestern Medical Center, Dallas, United States; [4]Howard Hughes Medical Institute, University of Texas Southwestern Medical Center, Dallas, United States

**Abstract** We previously discovered a new osteogenic growth factor that is required to maintain adult skeletal bone mass, Osteolectin/Clec11a. Osteolectin acts on Leptin Receptor[+] (LepR[+]) skeletal stem cells and other osteogenic progenitors in bone marrow to promote their differentiation into osteoblasts. Here we identify a receptor for Osteolectin, integrin $\alpha$11, which is expressed by LepR[+] cells and osteoblasts. $\alpha$11$\beta$1 integrin binds Osteolectin with nanomolar affinity and is required for the osteogenic response to Osteolectin. Deletion of *Itga11* (which encodes $\alpha$11) from mouse and human bone marrow stromal cells impaired osteogenic differentiation and blocked their response to Osteolectin. Like *Osteolectin* deficient mice, *Lepr-cre; Itga11[fl/fl]* mice appeared grossly normal but exhibited reduced osteogenesis and accelerated bone loss during adulthood. Osteolectin binding to $\alpha$11$\beta$1 promoted Wnt pathway activation, which was necessary for the osteogenic response to Osteolectin. This reveals a new mechanism for maintenance of adult bone mass: Wnt pathway activation by Osteolectin/$\alpha$11$\beta$1 signaling.
DOI: https://doi.org/10.7554/eLife.42274.001

*For correspondence:
sean.morrison@utsouthwestern.edu

## Introduction

The maintenance of the adult skeleton requires the formation of new bone throughout life as a result of the differentiation of skeletal stem/progenitor cells into osteoblasts. Leptin Receptor[+] (LepR[+]) bone marrow stromal cells are the major source of osteoblasts and adipocytes in adult mouse bone marrow (*Zhou et al., 2014*). These cells arise postnatally in the bone marrow, where they are initially rare and make little contribution to the skeleton during development, but expand to account for 0.3% of cells in adult bone marrow (*Mizoguchi et al., 2014*; *Zhou et al., 2014*). Nearly all fibroblast colony-forming cells (CFU-F) in adult mouse bone marrow arise from these LepR[+] cells, and a subset of LepR[+] cells form multilineage colonies containing osteoblasts, adipocytes, and chondrocytes, suggesting they are highly enriched for skeletal stem cells (*Zhou et al., 2014*). LepR[+] cells are also a critical source of growth factors that maintain hematopoietic stem cells and other primitive hematopoietic progenitors in bone marrow (*Ding and Morrison, 2013*; *Ding et al., 2012*; *Himburg et al., 2018*; *Oguro et al., 2013*).

To identify new growth factors in the bone marrow, we performed RNA-seq analysis on LepR[+] cells and looked for transcripts predicted to encode secreted proteins with sizes and structures similar to growth factors and whose function had not been studied in vivo. We discovered that Clec11a,

**eLife digest** Throughout our lives, our bones undergo constant remodeling. Cells called osteoclasts break down old bone and cells called osteoblasts lay down new. Normally, the two cell types work in balance but if the rate of breakdown outpaces new bone formation the skeleton can become weak. This weakness leads to a condition called osteoporosis, in which people suffer from fragile bones. Osteoporosis is hard to reverse, in part because our ability to encourage new bone to form is limited.

In 2016, researchers discovered a protein called osteolectin, which promotes new bone formation during adulthood by helping skeletal stem cells transform into bone cells. But so far, it has been unclear how osteolectin achieves this. To investigate this further, Shen et al. – including some researchers involved in the 2016 study – marked osteolectin with a molecular tag and tested what it bound on the surface of mouse and human bone marrow cells.

The experiments revealed that osteolectin binds to a specific receptor protein called α11 integrin, which can only be found on skeletal stem cells and the osteoblasts they give rise to. Once osteolectin binds to the receptor, it activates a signaling pathway that induces the stem cells to develop into osteoblasts. Mice that lacked either osteolectin or α11 integrin produced less bone and lost bone tissue faster as adults.

Osteolectin could potentially be useful in the treatment of osteoporosis or broken bones. Since only skeletal stem cells and osteoblasts cells produce α11 integrin, osteolectin would specifically target these cells without affecting cells that do not form bones. A next step will be to assess how well osteolectin compares to existing treatments for fragile bones.

DOI: https://doi.org/10.7554/eLife.42274.002

a secreted glycoprotein of the C-type lectin domain superfamily (*Bannwarth et al., 1999*; *Bannwarth et al., 1998*), was preferentially expressed by LepR$^+$ cells (*Yue et al., 2016*). Prior studies had observed *Clec11a* expression in bone marrow but inferred based on colony-forming assays in culture that it was a hematopoietic growth factor (*Hiraoka et al., 1997*; *Hiraoka et al., 2001*). We made germline knockout mice and found it is not required for normal hematopoiesis but that it is required for the maintenance of the adult skeleton (*Yue et al., 2016*). The mutant mice formed their skeleton normally during development and were otherwise grossly normal as adults but exhibited significantly reduced osteogenesis and bone volume beginning by 2 months of age (*Yue et al., 2016*). Recombinant protein promoted osteogenic differentiation by bone marrow stromal cells in vitro and in vivo (*Yue et al., 2016*). Based on these observations we proposed to call this new osteogenic growth factor, Osteolectin, so as to have a name related to its biological function. Osteolectin/Clec11a is expressed by a subset of LepR$^+$ stromal cells in the bone marrow as well as by osteoblasts, osteocytes, and hypertrophic chondrocytes. The discovery of Osteolectin offers the opportunity to better understand the mechanisms that maintain the adult skeleton; however, the Osteolectin receptor and the signaling mechanisms by which it promotes osteogenesis are unknown.

Several families of growth factors, and the signaling pathways they activate, promote osteogenesis, including Bone Morphogenetic Proteins (BMPs), Fibroblast Growth Factors (FGFs), Hedgehog proteins, Insulin-Like Growth Factors (IGFs), Transforming Growth Factor-betas (TGF-βs), and Wnts (reviewed by *Karsenty, 2003*; *Kronenberg, 2003*; *Wu et al., 2016*). Bone marrow stromal cells regulate osteogenesis by skeletal stem/progenitor cells by secreting multiple members of these growth factor families (*Chan et al., 2015*). The Wnt signaling pathway is a particularly important regulator of osteogenesis, as GSK3 inhibition and β-catenin accumulation promote the differentiation of skeletal stem/progenitor cells into osteoblasts (*Bennett et al., 2005*; *Dy et al., 2012*; *Hernandez et al., 2010*; *Krishnan et al., 2006*; *Kulkarni et al., 2006*; *Rodda and McMahon, 2006*). Consistent with this, mutations that promote Wnt pathway activation increase bone mass in humans and in mice (*Ai et al., 2005*; *Balemans et al., 2001*; *Boyden et al., 2002*) while mutations that reduce Wnt pathway activation reduce bone mass in humans and in mice (*Gong et al., 2001*; *Holmen et al., 2004*; *Kato et al., 2002*).

The Wnt pathway can be activated by integrin signaling. There are 18 integrin α subunits and 8 β subunits, forming 24 different functional integrin heterodimer complexes (*Humphries et al., 2006*; *Hynes, 1992*). Integrin signaling promotes Wnt pathway activation through Integrin-Linked Kinase (ILK)-mediated phosphorylation of GSK3 and nuclear translocation of β-catenin (*Burkhalter et al., 2011*; *Delcommenne et al., 1998*; *Novak et al., 1998*; *Rallis et al., 2010*). Conditional deletion of *Ilk* or *Ptk2* (which encodes Focal Adhesion Kinase, FAK) from osteoblast progenitors reduces osteogenesis and depletes trabecular bone in adult mice (*Dejaeger et al., 2017*; *Sun et al., 2016*), suggesting a role for integrins in adult osteogenesis. Conditional deletion of β1 integrin from chondrocytes or skeletal stem/progenitor cells impairs chondrocyte function and skeletal ossification during development (*Aszodi et al., 2003*; *Raducanu et al., 2009*; *Shekaran et al., 2014*). Activation of αvβ1 signaling by Osteopontin (*Chen et al., 2014*) or α5β1 signaling by Fibronectin (*Hamidouche et al., 2009*; *Moursi et al., 1997*) promotes the osteogenic differentiation of mesenchymal progenitors. Germline deletion of integrin α10 leads to defects in chondrocyte proliferation and growth plate function (*Bengtsson et al., 2005*) and germline deletion of integrin α11 leads to defects in tooth development (*Popova et al., 2007*). However, little is known about which integrins are required for adult osteogenesis in vivo.

## Results

### Integrin α11 is selectively expressed by LepR +cells and osteoblasts

The *Osteolectin/Clec11a* gene first appeared in bony fish and is conserved among bony vertebrates (*Yue et al., 2016*). Osteolectin contains a glutamic acid-rich sequence, an alpha-helical leucine zipper, and a C-type lectin domain (*Figure 1A and B*) (*Bannwarth et al., 1999*; *Bannwarth et al., 1998*). To generate hypotheses regarding potential Osteolectin receptors, we examined the amino acid sequence and found two integrin-binding motifs, RGD (*Gardner and Hynes, 1985*; *Pierschbacher and Ruoslahti, 1984*; *Plow et al., 1985*) and LDT (*Fong et al., 1997*; *Viney et al., 1996*) in human (*Figure 1A*) and mouse Osteolectin (*Figure 1B*). One or both of these motifs were conserved across Osteolectin sequences in all bony vertebrates (*Figure 1C*). This raised the possibility that the Osteolectin receptor might be an integrin.

Given that bone marrow stromal cells undergo osteogenesis in response to Osteolectin (*Yue et al., 2016*), we examined the expression of all α and β integrins in mouse bone marrow stromal cells by RNA-seq analysis. Among the genes that encode α integrins, *Itga1* (encoding α1), *Itga6* (encoding α6), *Itga11* (encoding α11) and *Itgav* (encoding αv), were strongly expressed by bone marrow stromal cells (*Figure 1D*). Among the genes that encode β integrins, only *Itgb1* (encoding β1) was strongly expressed (*Figure 1E*). *Itga1*, *Itga6*, and *Itgav* were strongly expressed by both LepR$^+$ cells and endothelial cells, and and are widely expressed in non-osteogenic cells, where they have known ligands (*Belkin et al., 1990*; *Defilippi et al., 1991*; *Lee et al., 2006*; *Mahabeleshwar et al., 2006*; *Yang et al., 2008*), suggesting they are less likely to encode the Osteolectin receptor (*Figure 1F*). In contrast, *Itga11* was expressed exclusively by LepR$^+$ cells, not by endothelial cells or other bone marrow cells (*Figure 1F*). Quantitative reverse transcription PCR (qRT-PCR) analysis of sorted bone marrow cells (*Supplementary file 1* shows the markers used to isolate these cells) showed that *Itga11* was highly expressed by LepR$^+$CD45$^-$Ter119$^-$CD31$^-$ stromal cells and *Col2.3-GFP$^+$*CD45$^-$Ter119$^-$CD31$^-$ osteoblasts but not by any hematopoietic stem or progenitor population (*Figure 1G*). The expression patterns of *Itga11* and *Itgb1* were thus consistent with a potential role in osteogenesis.

Consistent with our results, integrin α11 is expressed by human bone marrow stromal cells in a way that correlates with osteogenic potential in culture (*Kaltz et al., 2010*); however, α11 is not known to regulate osteogenesis. Integrin α11 heterodimerizes with integrin β1 (*Velling et al., 1999*) and the only known ligand for α11β1 is collagen (*Popova et al., 2004*; *Velling et al., 1999*). Few cells express integrin α11, and it has been studied less than other integrins. *Itga11* deficient mice are growth retarded and have smaller bones, but this was thought to reflect a defect in incisor development that leads to malnutrition (*Popova et al., 2007*).

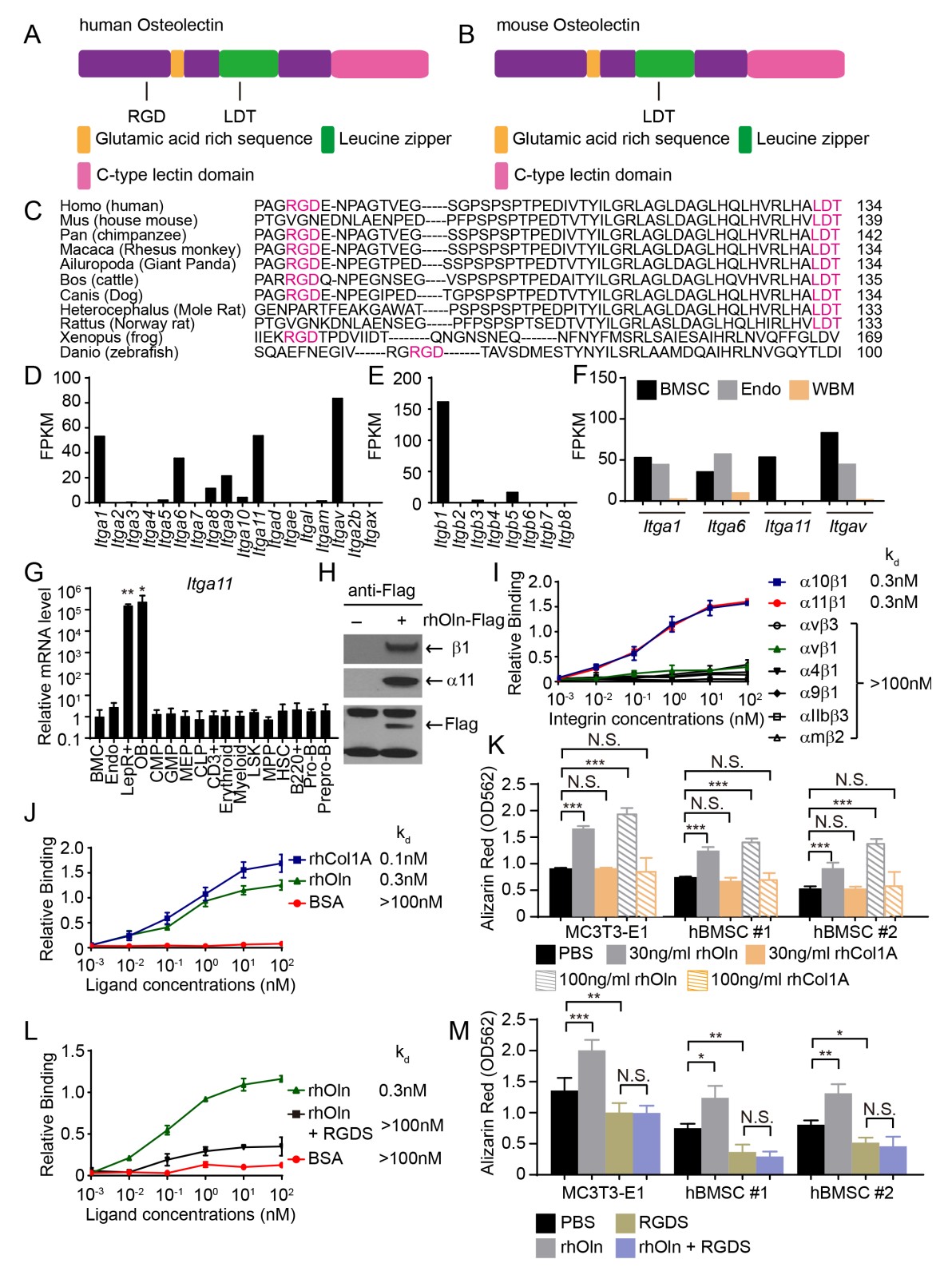

**Figure 1.** Osteolectin contains conserved integrin binding motifs and binds with high affinity to integrin $\alpha_{11}\beta_1$. (**A, B**) The human (**A**) and mouse (**B**) Osteolectin proteins contain RGD and LDT sequences. (**C**) Alignment of Osteolectin amino acid sequences shows that the RGD and LDT domains are evolutionarily conserved among bony vertebrates. (**D, E**) RNA-seq analysis of integrin $\alpha$ (**D**) and $\beta$ (**E**) subunits in PDGFR$\alpha^+$CD45$^-$Ter119$^-$CD31$^-$ bone marrow stromal cells from enzymatically dissociated adult bone marrow (n = 2 independent samples). These cells are uniformly positive for LepR

*Figure 1 continued on next page*

*Figure 1 continued*

expression (*Zhou et al., 2014*). (F) RNA-seq analysis of *Itga1, Itga6, Itga11,* and *Itgav* in PDGFRα⁺CD45⁻Ter119⁻CD31⁻ bone marrow stromal cells, VE-Cadherin⁺ bone marrow endothelial cells, and whole bone marrow cells (n = 2 independent samples per cell population). (G) *Itga11* expression in cell populations from mouse bone marrow by qRT-PCR (n = 3 independent samples per cell population). The markers used for the isolation of each cell population are shown in **Supplementary file 1**. (H) In MC3T3-E1 preosteoblast cells expressing Flag-tagged Osteolectin, anti-Flag antibody co-immunoprecipitated endogenous integrin β1 and integrin α11 with Flag-tagged Osteolectin (results are representative of two independent experiments). (I) Recombinant human Osteolectin (rhOln) selectively bound to recombinant human integrin $\alpha_{11}\beta_1$ and $\alpha_{10}\beta_1$, but not to other integrins (n = 3 independent experiments). (J) Integrin α11β1 bound Osteolectin and recombinant human Pro-Collagen 1α (rhCol1A) with similar affinities, but not bovine serum albumin (BSA) (n = 3 independent experiments). (K) Osteolectin, but not Pro-Collagen 1α, promoted osteogenic differentiation by MC3T3-E1 cells and human bone marrow stromal cells (n = 3 independent experiments). (L) 200 nM RGDS peptide inhibited the binding of integrin α11β1 to recombinant human Osteolectin. (M) 100 μM RGDS peptide inhibited osteogenic differentiation by MC3T3-E1 cells and human bone marrow stromal cells in response to 30 ng/ml of recombinant human Osteolectin. All numerical data reflect mean ±standard deviation. Statistical significance was determined with one-way (G) or two-way ANOVAs with Dunnett's multiple comparisons tests (K) or Tukey's multiple comparisons tests (M).
DOI: https://doi.org/10.7554/eLife.42274.003

The following source data is available for figure 1:

**Source data 1.** Data for *Figure 1*.
DOI: https://doi.org/10.7554/eLife.42274.004

## Osteolectin binds to α11β1 with nanomolar affinity

To test whether α11β1 binds Osteolectin, we overexpressed Flag-tagged human Osteolectin in MC3T3-E1 pre-osteoblast cells and immunoprecipitated with anti-Flag beads. The anti-Flag beads pulled down the tagged Osteolectin along with endogenous integrin α11 and integrin β1 (*Figure 1H*). We then tested the affinity of recombinant human Osteolectin for multiple recombinant human integrin complexes by a microtiter well binding assay. Osteolectin selectively bound to integrin α11β1 and α10β1, but not to other integrins, including αVβ1, αVβ3, α4β1, α9β1, αIIbβ3 or αMβ2 (*Figure 1I*). Integrin α10 is the gene most closely related to α11. Integrin α10 is expressed by osteoblasts and chondrocytes (*Bengtsson et al., 2005*; *Engel et al., 2013*) but only at a low level by bone marrow stromal cells (*Figure 1D*). The dissociation constant ($k_d$) of Osteolectin for α10β1 and α11β1 was 0.3 nM whereas the $k_d$ of Osteolectin for other integrins was >100 nM. The $k_d$ of human Pro-Collagen 1α for α11β1 was also high (0.1 nM; *Figure 1J*); however, in contrast to Osteolectin, addition of Pro-Collagen 1α to culture, either by adding it to the culture medium or using it to coat the plates, did not promote osteogenesis by MC3T3-E1 cells or two primary human bone marrow stromal cell lines (hBMSC#1 or hBMSC#2; *Figure 1K*). A peptide that inhibits the binding of integrins to RGD-containing ligands, RGDS (Arg-Gly-Asp-Ser) (*Gardner and Hynes, 1985*; *Plow et al., 1985*; *Ruoslahti, 1996*), inhibited the binding of integrin α11β1 to Osteolectin (*Figure 1L*) and the osteogenic response of MC3T3-E1 cells, hBMSC#1 cells, and hBMSC#2 cells to Osteolectin in culture (*Figure 1M*).

## Osteolectin promotes Wnt pathway activation

MC3T3-E1 cells, hBMSC#1 cells, and hBMSC#2 cells secrete Osteolectin into the culture medium (*Figure 2A*), consistent with our observation that Osteolectin is synthesized by a subset of LepR⁺ bone marrow stromal cells (*Yue et al., 2016*). Deletion of *Osteolectin* from these cell lines reduced their osteogenic differentiation in osteogenic differentiation medium (*Figure 2B and C*), demonstrating that autocrine Osteolectin production is part of what drives osteogenesis by these cells in culture.

To assess the signaling mechanisms by which Osteolectin promotes osteogenesis, we treated parental MC3T3-E1 cells, hBMSC#1 cells, and hBMSC#2 cells with recombinant Osteolectin and assessed the levels of phosphorylated PI3-kinase, Akt, and GSK3. The most prominent change we observed was a dramatic increase of GSK3 phosphorylation at Ser21/9 within 30 to 60 min of Osteolectin treatment in all three cell lines (*Figure 2D*). Phosphorylation at Ser21/9 inhibits the GSK3-mediated degradation of β-catenin, increasing β-catenin levels and promoting the transcription of Wnt pathway target genes (*Cross et al., 1995*; *Peifer et al., 1994*; *Yost et al., 1996*). We did not observe an increase in β-catenin levels within 1 hr of Osteolectin treatment, but did detect increased β-catenin levels in all three cell lines within 24 hr of Osteolectin treatment (*Figure 2E*). The transcription of several Wnt target genes, including *Axin2* (*Jho et al., 2002*; *Lustig et al., 2002*; *Yan et al.,*

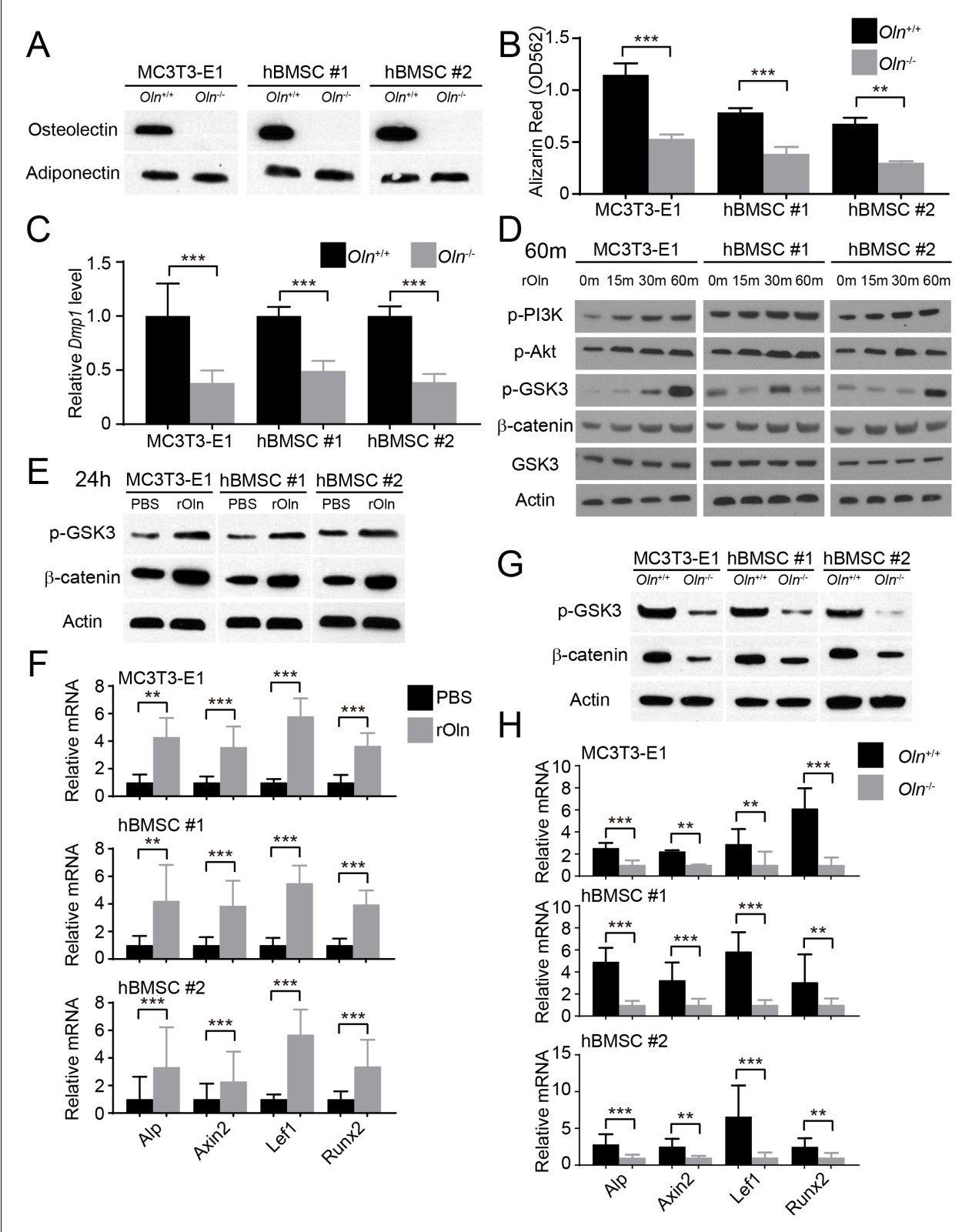

**Figure 2.** Osteolectin activates Wnt pathway signaling in skeletal stem/progenitor cells. (**A**) Western blot of cell culture supernatant from parental or *Osteolectin* deficient MC3T3-E1 cells and human bone marrow stromal cells (hBMSC#1 and hBMSC#2 cells) (this blot is representative of blots from three independent experiments). (**B**) Osteogenic differentiation in culture of parental or *Osteolectin* deficient MC3T3-E1 cells, hBMSC#1 cells, and hBMSC#2 cells. Alizarin red staining was performed after 14 days (for MC3T3-E1 cells) or 21 days (for hBMSC cells) to quantify osteoblast differentiation

*Figure 2 continued on next page*

*Figure 2 continued*

and mineralization (n = 3 independent experiments). (**C**) qRT-PCR analysis of *Dmp1* transcript levels in MC3T3-E1 cells, hBMSC#1 cells, and hBMSC#2 cells after 14 to 21 days of osteogenic differentiation (n = 5 independent experiments). (**D**) MC3T3-E1 cells, hBMSC#1 cells, and hBMSC#2 cells were stimulated with recombinant mouse Osteolectin (rOln) in osteogenic differentiation medium then lysed 0, 15, 30, or 60 min later and immunoblotted for phospho-PI3K, phospho-Akt, phospho-GSK3, β-catenin, total GSK3, and Actin (results are representative of 3 independent experiments). (**E**) MC3T3-E1 cells, hBMSC#1 cells, and hBMSC#2 cells were transferred into osteogenic differentiation medium for 24 hr with PBS or recombinant mouse Osteolectin, lysed, and immunoblotted for phospho-GSK3, β-catenin, and Actin (this blot is representative of blots from three independent experiments). (**F**) qRT-PCR analysis of Wnt target gene transcript levels (*Alkaline phosphatase, Axin2, Lef1,* or *Runx2*) in MC3T3-E1 cells, hBMSC#1 cells, and hBMSC#2 cells 24 hr after transfer into osteogenic differentiation medium, with PBS or recombinant mouse Osteolectin (n = 5 independent experiments). (**G**) Parental or *Osteolectin* deficient MC3T3-E1 cells, hBMSC#1 cells, and hBMSC#2 cells after 24 hr in osteogenic differentiation medium were lysed and immunoblotted for phospho-GSK3, β-catenin, and Actin (this blot is representative of blots from three independent experiments). (**H**) qRT-PCR analysis of Wnt target gene transcript levels in parental or *Osteolectin* deficient MC3T3-E1 cells, hBMSC#1 cells, and hBMSC#2 cells 24 hr after transfer into osteogenic differentiation medium (n = 5 independent experiments). All numerical data reflect mean ±standard deviation. The statistical significance of differences was determined with two-way ANOVAs with Sidak's multiple comparisons tests.

DOI: https://doi.org/10.7554/eLife.42274.005

The following source data is available for figure 2:

**Source data 1.** Data for *Figure 2*.
DOI: https://doi.org/10.7554/eLife.42274.006

*2001*; *Yan et al., 2009*), *Lef1* (*Filali et al., 2002*; *Gaur et al., 2005*; *Hovanes et al., 2001*), *Runx2* (*Dong et al., 2006*; *Gaur et al., 2005*), and *Alkaline phosphatase* (*Rawadi et al., 2003*), were activated within 24 hr of Osteolectin treatment (*Figure 2F*). *Osteolectin* deficient MC3T3-E1 cells, hBMSC#1 cells, and hBMSC#2 cells had lower levels of phospho-GSK3 and β-catenin as compared to parental cells (*Figure 2G*) as well as significantly lower levels of Wnt target genes (*Figure 2H*). These data demonstrate that Osteolectin promotes Wnt pathway activation in osteogenic cells.

## Osteogenesis in response to Osteolectin requires β-catenin

To test whether Wnt pathway activation phenocopies the effects of Osteolectin, we evaluated the effects of AZD2858, which inhibits GSK3 function and promotes β-catenin accumulation (*Berg et al., 2012*). As expected (*Gilmour et al., 2013*; *Marsell et al., 2012*; *Sisask et al., 2013*), AZD2858 increased GSK3 phosphorylation and β-catenin levels (*Figure 3A*) as well as osteogenic differentiation (*Figure 3B*) by MC3T3-E1 cells, hBMSC#1 cells, and hBMSC#2 cells. Osteolectin also increased GSK3 phosphorylation, β-catenin levels, and osteogenic differentiation by MC3T3-E1 cells, hBMSC#1 cells, and hBMSC#2 cells (*Figure 3A and B*). However, when the two agents were added together, there was no further promotion of osteogenic differentiation beyond the effects of the individual agents (*Figure 3B*). These data suggest that Osteolectin and GSK3/β-catenin act in the same pathway to promote osteogenic differentiation by mesenchymal progenitors.

To test whether Osteolectin requires β-catenin to promote osteogenesis, we evaluated an inhibitor of Wnt pathway signaling, IWR-1-endo, which depletes β-catenin by stabilizing Axin2 in the β-catenin destruction complex (*Chen et al., 2009*). As expected, Osteolectin increased β-catenin levels and osteogenesis by MC3T3-E1 cells, hBMSC#1 cells, and hBMSC#2 cells while IWR-1-endo reduced β-catenin levels and osteogenesis (*Figure 3C and D*). When added together, IWR-1-endo blocked the effect of Osteolectin on β-catenin levels and osteogenesis (*Figure 3C and D*). This suggests that Osteolectin requires β-catenin to induce osteogenesis by mesenchymal progenitors.

## Wnt pathway activation and osteogenesis by Osteolectin require integrin α11

To test if Integrin α11 is required for the osteogenic response to Osteolectin, we used CRISPR/Cas9 to delete *Itga11* from MC3T3-E1 cells, hBMSC#1 cells, and hBMSC#2 cells. Deletion of *Itga11* significantly reduced osteogenesis by each cell line in culture (*Figure 4A and B*). Addition of recombinant Osteolectin to culture significantly promoted osteogenesis by parental, but not *Itga11* deficient, MC3T3-E1, hBMSC#1, and hBMSC#2 cells (*Figure 4A and B*). Integrin α11 is therefore required by mouse pre-osteoblast cells and human bone marrow stromal cells to undergo osteogenesis in response to Osteolectin.

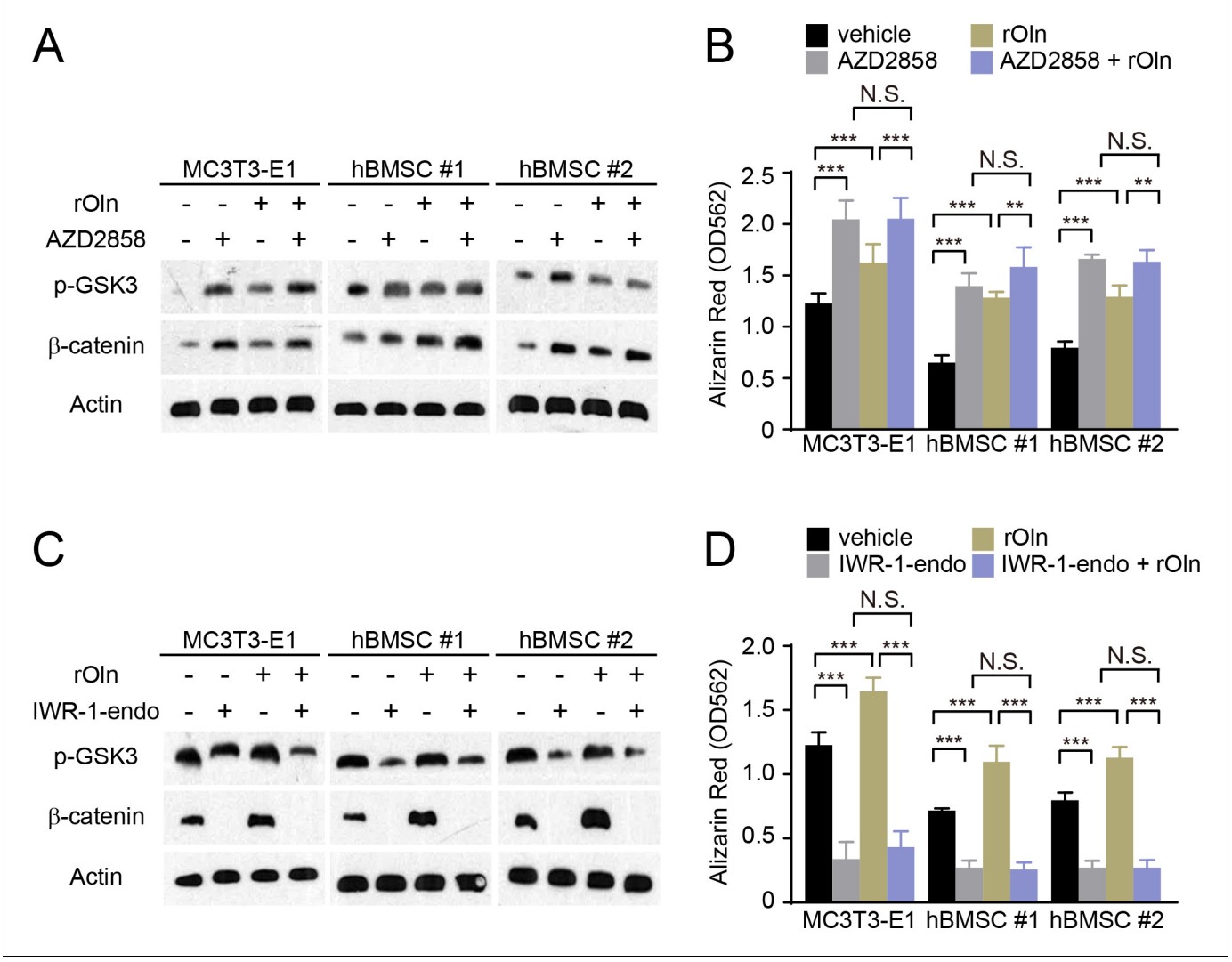

**Figure 3.** Osteogenic differentiation in response to Osteolectin requires β-catenin. (**A and B**) MC3T3-E1 cells, hBMSC#1 cells, and hBMSC#2 cells were transferred into osteogenic differentiation medium with PBS or 30 ng/ml recombinant mouse Osteolectin, as well as DMSO or 200 nM of the GSK3 inhibitor AZD2858. (**A**) Cells were lysed 24 hr later and immunoblotted for phospho-GSK3, β-catenin, and Actin. (**B**) Alizarin red staining after 14 days (MC3T3-E1 cells) or 21 days (hBMSC cells) to quantify osteoblast differentiation and mineralization (n = 3 independent experiments). (**C and D**) MC3T3-E1 cells, hBMSC#1 cells, and hBMSC#2 cells were transferred into osteogenic differentiation medium with PBS or 30 ng/ml Osteolectin, as well as DMSO or 200 nM of the β-catenin inhibitor IWR-1-endo. (**C**) Cells were lysed 24 hr later and immunoblotted for phospho-GSK3, β-catenin, and Actin. (**D**) Alizarin red staining after 14 days (MC3T3-E1 cells) or 21 days (hBMSC cells) to quantify osteoblast differentiation and mineralization (n = 3 independent experiments). All numerical data reflect mean ±standard deviation. The statistical significance of differences was determined with two-way ANOVAs with Tukey's multiple comparisons tests.

DOI: https://doi.org/10.7554/eLife.42274.007

The following source data is available for figure 3:

**Source data 1.** Data for Data for *Figure 3*.
DOI: https://doi.org/10.7554/eLife.42274.008

The *Itga11* deficient MC3T3-E1 cells, hBMSC#1 cells, and hBMSC#2 cells also exhibited lower levels of GSK3 phosphorylation and β-catenin as compared to parental control cells (*Figure 4C*). Addition of recombinant Osteolectin to culture increased the levels of phosphorylated GSK3 and β-catenin in parental control cells but not in *Itga11* deficient cells (*Figure 4C*). We also observed significantly lower levels of Wnt target gene transcripts in the *Itga11* deficient cells (*Figure 4D*). Addition of recombinant Osteolectin to culture significantly increased the levels of Wnt target gene

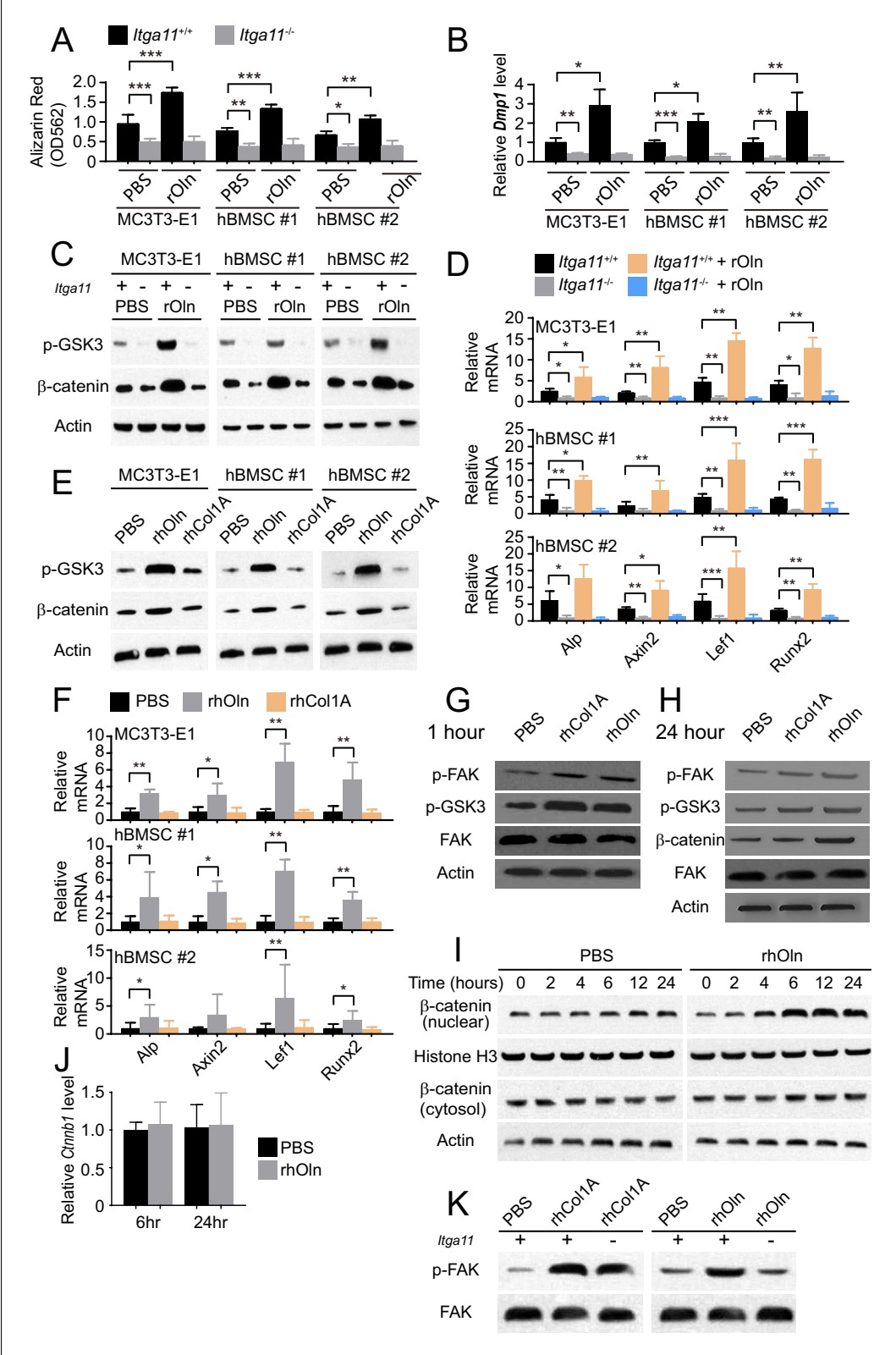

**Figure 4.** Integrin α11 is required for osteogenic differentiation and Wnt pathway activation in response to Osteolectin but exogenous collagen does not activate Wnt pathway signaling. (**A**) Osteogenic differentiation in culture of parental or *Itga11* deficient MC3T3-E1 cells, hBMSC#1 cells, and hBMSC#2 cells with PBS or recombinant mouse Osteolectin (n = 3 independent experiments). (**B**) qRT-PCR analysis of *Dmp1* transcript levels in MC3T3-E1 cells, hBMSC#1 cells, and hBMSC#2 cells after 14 to 21 days of osteogenic differentiation (n = 5 independent experiments). (**C**) Parental or *Itga11*

*Figure 4 continued on next page*

*Figure 4 continued*

deficient MC3T3-E1 cells, hBMSC#1 cells, and hBMSC#2 cells were transferred into osteogenic differentiation medium with or without recombinant mouse Osteolectin then lysed 24 hr later and immunoblotted for phospho-GSK3, β-catenin, and Actin (this blot is representative of blots from three independent experiments). (D) qRT-PCR analysis of Wnt target gene transcript levels in parental or *Itga11* deficient MC3T3-E1 cells, hBMSC#1 cells, and hBMSC#2 cells 24 hr after transfer into osteogenic differentiation medium with PBS or Osteolectin (n = 5 independent experiments). (E) MC3T3-E1 cells, hBMSC#1 cells, and hBMSC#2 cells were transferred into osteogenic differentiation medium with PBS or 30 ng/ml Osteolectin, or 30 ng/ml recombinant Pro-Collagen 1α, then lysed 24 hr later and immunoblotted for phospho-GSK3, β-catenin, and Actin (this blot is representative of blots from three independent experiments). (F) qRT-PCR analysis of Wnt target gene transcript levels in MC3T3-E1 cells, hBMSC#1 cells, and hBMSC#2 cells 24 hr after transfer into osteogenic differentiation medium with PBS or 30 ng/ml Osteolectin, or 30 ng/ml Pro-Collagen 1α (n = 5 independent experiments). (G) Primary mouse bone marrow stromal cells were adherently cultured in osteogenic differentiation medium. PBS (control), recombinant human Pro-Collagen 1α, or recombinant human Osteolectin was added, then the cells were lysed 1 hr later and lysates were immunoblotted for phospho-FAK, phospho-GSK3, total FAK, and Actin (this blot is representative of blots from three independent experiments). (H) Primary mouse bone marrow stromal cells adherently cultured in osteogenic differentiation medium were treated with PBS (control), recombinant human Pro-Collagen 1α, or recombinant human Osteolectin then lysed 24 hr later and lysates were immunoblotted for phospho-FAK, phospho-GSK3, β-catenin, total FAK, and Actin (this blot is representative of blots from three independent experiments). (I) Primary mouse bone marrow stromal cells adherently cultured in osteogenic differentiation medium were treated with 30 ng/ml recombinant human Osteolectin or PBS (control) then lysed 2, 4, 6, 12, or 24 hr later. Nuclear and cytosolic/membrane-associated fractions were isolated from lysates by centrifugation then immunoblotted for β-catenin. As loading controls, Histone H3 was blotted in the nuclear fraction and Actin was blotted in the cytosolic/membrane-associated fraction (this blot is representative of blots from three independent experiments). (J) qRT-PCR analysis of *Ctnnb1* transcript levels in cells from the experiment in panel (I) (n = 3 independent experiments). (K) Primary mouse bone marrow stromal cells from *Lepr-Cre; Itga11*$^{fl/fl}$ or littermate control mice were adherently cultured in osteogenic differentiation medium. PBS (control), recombinant human Pro-Collagen 1α, or recombinant human Osteolectin was added then the cells were lysed 1 hr later and lysates were immunoblotted for phospho-FAK and FAK (this blot is representative of blots from two independent experiments). All numerical data reflect mean ±standard deviation. Statistical significance was determined with two-way ANOVAs with Tukey's multiple comparisons tests (A, B and D), Dunnett's multiple comparisons tests (F), or Sidak's multiple comparisons tests (J).

DOI: https://doi.org/10.7554/eLife.42274.009

The following source data is available for figure 4:

**Source data 1.** Data for *Figure 4*.
DOI: https://doi.org/10.7554/eLife.42274.010

transcripts in parental control cells but not in *Itga11* deficient cells (*Figure 4D*). Integrin α11 is therefore required by mouse pre-osteoblast cells and human bone marrow stromal cells to activate Wnt pathway signaling in response to Osteolectin. Given that Collagen 1α bound to α11β1 (*Figure 1J*) but did not promote osteogenesis by MC3T3-E1 cells, hBMSC#1 cells, or hBMSC#2 cells (*Figure 1K*), we tested whether Collagen 1α promoted Wnt pathway activation in these cells. Addition of recombinant Osteolectin to culture increased levels of phosphorylated GSK3, β-catenin, and Wnt target gene transcripts in MC3T3-E1 cells, hBMSC#1 cells, and hBMSC#2 cells (*Figure 4E and F*); however, addition of Pro-Collagen 1α to these cells did not seem to have any effect on the levels of phosphorylated GSK3, β-catenin, or Wnt target gene transcripts in these cells (*Figure 4E and F*).

We also added Osteolectin or Pro-Collagen 1α to freshly isolated mouse bone marrow stromal cells in culture. Both Pro-Collagen 1α and Osteolectin promoted Focal Adhesion Kinase (FAK) phosphorylation at Tyrosine 397 within one hour of addition to culture without affecting total FAK levels (*Figure 4G*), consistent with the activation of integrin signaling (*Cooper et al., 2003*). Although Pro-Collagen 1α had not detectably affected GSK3 phosphorylation in MC3T3-E1 cells, hBMSC#1 cells, or hBMSC#2 cells (*Figure 4E*), it did increase the levels of phosphorylated GSK3 in primary mouse bone marrow stromal cells, as did Osteolectin (*Figure 4G and H*). Osteolectin, but not Pro-Collagen 1α, promoted β-catenin accumulation within 24 hr of treatment (*Figure 4H*). To more precisely assess the time required for Osteolectin to promote β-catenin accumulation, we treated primary mouse bone marrow stromal cells with Osteolectin or phosphate-buffered saline (PBS) then assessed the levels of nuclear versus cytosolic/membrane associated β-catenin 2, 4, 6, 12, and 24 hr later. Osteolectin treatment did not significantly affect the levels of cytosolic/membrane associated β-catenin at any time point, but did increase nuclear β-catenin within 4 hr of treatment (*Figure 4I*). Osteolectin treatment did not significantly affect *Ctnnb1* (which encodes β-catenin) transcript levels in these cells (*Figure 4J*), suggesting that the increase in nuclear β-catenin reflected an inhibition of proteasomal degradation. Together, these data suggest that Osteolectin promoted integrin α11 signaling, leading to Wnt pathway activation, accumulation of nuclear β-catenin, and increased transcription of Wnt target genes involved in osteogenesis.

Exogenous Pro-Collagen 1α also appeared to activate integrin signaling, at least in primary mouse bone marrow stromal cells (*Figure 4G*), but not accumulation of nuclear β-catenin (*Figure 4H*) or increased transcription of Wnt target genes (*Figure 4F*), offering a potential explanation for its failure to promote osteogenesis. *Itga11*-deficiency blocked the ability of Osteolectin to promote FAK phosphorylation in primary bone marrow stromal cells, but did not affect the ability of Pro-Collagen 1α to promote FAK phosphorylation (*Figure 4K*). This suggests that Osteolectin promotes integrin signaling in an α11-dependent manner but that Pro-Collagen 1α promotes integrin signaling in an α11-independent manner. This was expected as bone marrow stromal cells express multiple integrins that are capable of functioning as collagen receptors, including α1β1, α10β1 and αVβ3 (*Figure 1D and E*) (*Davis, 1992*; *Gullberg and Lundgren-Akerlund, 2002*).

## Conditional deletion of integrin α11 from LepR$^+$ cells reduces Osteogenesis in vivo

To test whether Integrin α11 is necessary for osteogenesis in vivo we generated mice bearing a floxed allele of *Itga11* (*Figure 5—figure supplement 1A–C*), then conditionally deleted it from skeletal stem and progenitor cells in the bone marrow using *Lepr-Cre*. Only 5% of osteoblasts derive from LepR$^+$ cells at two months of age but this number increases to approximately 50% by 10 months of age (*Zhou et al., 2014*). *Lepr-Cre; Itga11*$^{fl/fl}$ mice did not exhibit the defects in incisor development (data not shown) or the growth retardation observed in germline *Itga11*$^{-/-}$ mice (*Popova et al., 2007*). *Lepr-Cre; Itga11*$^{fl/fl}$ mice appeared grossly normal (*Figure 5A*), with body lengths (*Figure 5B*), body masses (*Figure 5C*), and femur lengths (*Figure 5D*) that did not significantly differ from sex-matched littermate controls. However, qRT-PCR analysis showed that LepR$^+$ bone marrow cells from *Lepr-Cre; Itga11*$^{fl/fl}$ mice had an approximately 85% reduction in *Itga11* transcript levels as compared to LepR$^+$ cells from control mice (*Figure 5—figure supplement 1D*). Serum Osteolectin levels did not significantly differ between *Lepr-Cre; Itga11*$^{fl/fl}$ mice and littermate controls at 2 or 12 months of age but were modestly higher in *Lepr-Cre; Itga11*$^{fl/fl}$ mice than in controls at 6 months of age (*Figure 5E*). This demonstrates that Integrin α11 is not required for the synthesis or secretion of Osteolectin.

To test whether deletion of *Itga11* from LepR$^+$ cells affected osteogenesis in vivo, we performed micro-CT analysis of the distal femur from 2, 6, and 12 month old *Lepr-Cre; Itga11*$^{fl/fl}$ mice and sex-matched littermates. Consistent with the observation that LepR$^+$ cells contribute little to skeletal development prior to 2 months of age (*Zhou et al., 2014*), we observed no significant difference in trabecular bone parameters between *Lepr-Cre; Itga11*$^{fl/fl}$ mice and sex-matched littermates at 2 months of age (*Figure 5F–L*). However, LepR$^+$ cells and Osteolectin are necessary for adult osteogenesis (*Yue et al., 2016*; *Zhou et al., 2014*). Consistent with this, 6 and 12-month-old male and female *Lepr-Cre; Itga11*$^{fl/fl}$ mice had significantly reduced trabecular bone volume as compared to sex-matched littermate controls (*Figure 5F and G*). At 6 and 12 months of age, male and female *Lepr-Cre; Itga11*$^{fl/fl}$ mice also tended to have lower trabecular number (*Figure 5H*) and trabecular thickness (*Figure 5I*) than sex matched littermate controls. Calcein double labelling showed that the mineral apposition rate was significantly reduced in trabecular bone from *Lepr-Cre; Itga11*$^{fl/fl}$ mice as compared to sex-matched littermates at 6 and 12 months of age (*Figure 5M*). Levels of Procollagen type 1 N-terminal Propeptide (P1NP), a marker of bone formation, were also significantly lower in the serum of *Lepr-Cre; Itga11*$^{fl/fl}$ mice as compared to littermate controls at 6 and 12 months of age (*Figure 5O*). Integrin α11 is, therefore, required by LepR$^+$ cells and their progeny for normal rates of trabecular bone formation and maintenance of trabecular bone volume during adulthood, phenocopying the accelerated trabecular bone loss in adult *Osteolectin* deficient mice (*Yue et al., 2016*).

While *Lepr-Cre; Itga11*$^{fl/fl}$ mice had significantly reduced rates of bone formation as compared to sex-matched littermates (*Figure 5M*), they did not significantly differ in the urinary bone resorption marker deoxypyridinoline at 6 or 12 months of age (*Figure 5N*; this was not tested in 2-month-old mice because no difference in bone parameters was observed at that age). This suggests that, like Osteolectin, Integrin α11 promotes bone formation but does not regulate bone resorption (*Yue et al., 2016*).

*Osteolectin* deficiency has a milder effect on cortical bone as compared to trabecular bone, with no significant reduction in cortical bone until after 10 months of age (*Yue et al., 2016*). Consistent with this, femur cortical bone parameters did not significantly differ between *Lepr-Cre; Itga11*$^{fl/fl}$ mice and sex-matched littermates at 2 or 6 month of age (*Figure 6A–F*). However, cortical bone

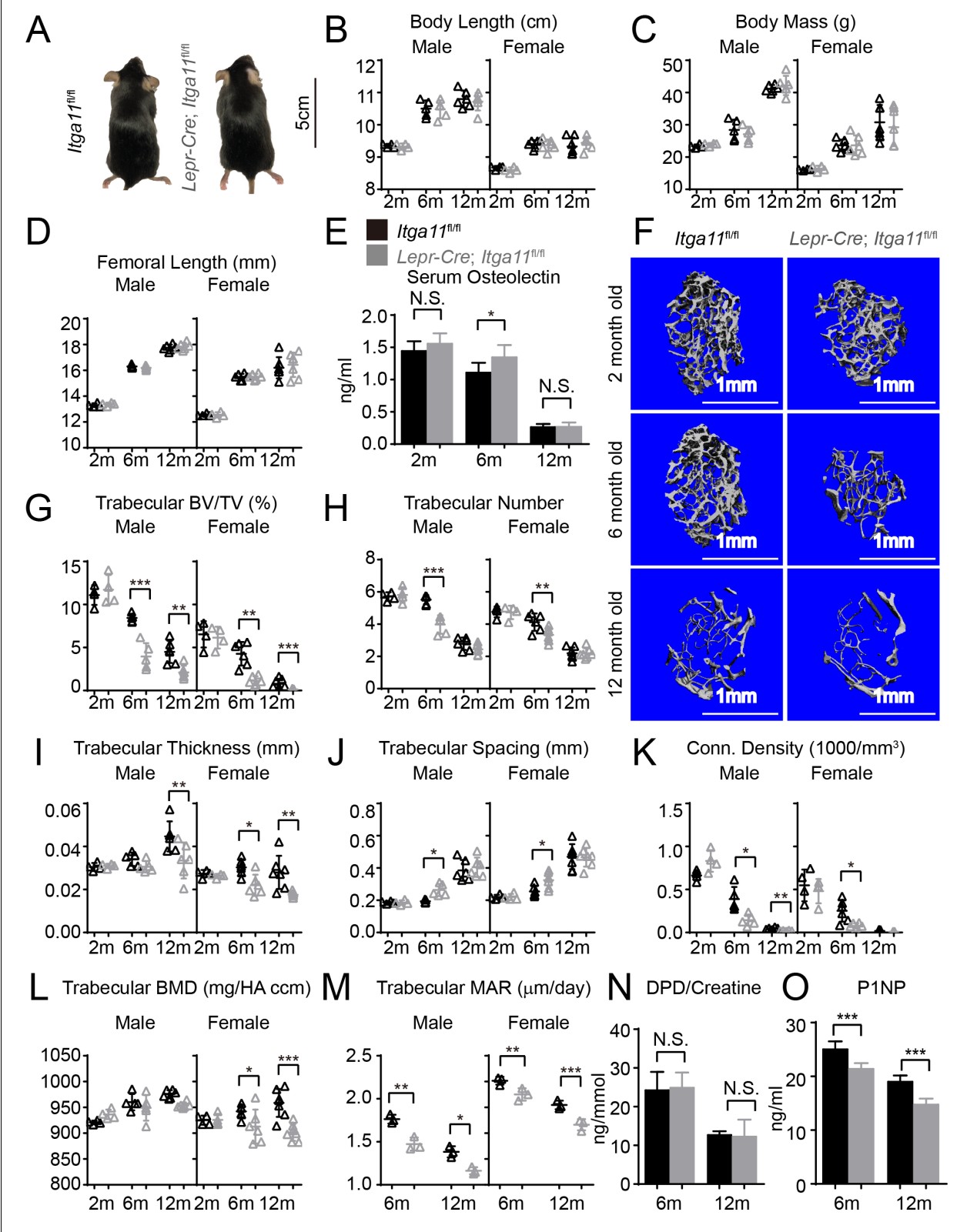

**Figure 5.** Conditional *Itga11* deletion from LepR+cells accelerates the loss of trabecular bone during aging. (A) *Lepr-Cre; Itga11fl/fl* mice were grossly normal and indistinguishable from littermate controls. (B – D) Body length (B), body mass (C) and femur length (D) did not significantly differ between *Lepr-Cre; Itga11fl/fl* mice and sex-matched littermate controls at 2, 6, or 12 months of age (n = 4–7 mice per genotype per sex per time point, from at least three independent experiments). (E) ELISA measurement of serum Osteolectin levels in *Lepr-Cre; Itga11fl/fl* mice and littermate controls at 2, 6,

*Figure 5 continued on next page*

*Figure 5 continued*

and 12 months of age (n = 8 mice per genotype per time point from four independent experiments). (F) Representative microCT images of trabecular bone in the distal femur metaphysis of male *Lepr-Cre; Itga11*<sup>fl/fl</sup> mice and littermate controls at 2, 6, and 12 months of age. (G–L) microCT analysis of trabecular bone volume/total volume (G), trabecular number (H), trabecular bone thickness (I), trabecular bone spacing (J), connectivity density (K), and bone mineral density (L) in the distal femur metaphysis of *Lepr-Cre; Itga11*<sup>fl/fl</sup> mice and sex-matched littermate controls at 2, 6, and 12 months of age (n = 4–7 mice per genotype per sex per time point from at least three independent experiments). (M) Trabecular bone mineral apposition rate based on calcein double labelling in the distal femur metaphysis (n = 3 mice per genotype per sex per time point). (N) Bone resorption rate analysis by measuring the deoxypyridinoline/creatinine ratio in the urine of *LepR-Cre; Itga11*<sup>fl/fl</sup> mice and littermate controls at 6 and 12 months of age (n = 4–5 mice per genotype per time point from three independent experiments). (O) Serum P1NP levels in *LepR-Cre; Itga11*<sup>fl/fl</sup> mice and sex-matched littermate controls at 6 and 12 months of age (n = 4 mice per genotype per time point from three independent experiments). All numerical data reflect mean ±standard deviation. The statistical significance of differences was determined with two-way ANOVAs with Sidak's multiple comparisons tests.
DOI: https://doi.org/10.7554/eLife.42274.011

The following source data and figure supplements are available for figure 5:

**Source data 1.** Data for Data for *Figure 5*.
DOI: https://doi.org/10.7554/eLife.42274.012
**Figure supplement 1.** Generation of an *Itga11* floxed mice.
DOI: https://doi.org/10.7554/eLife.42274.013
**Figure supplement 1—source data 1.** Data for *Figure 5—figure supplement 1*.
DOI: https://doi.org/10.7554/eLife.42274.014

mineral density was significantly lower in male and female *Lepr-Cre; Itga11*<sup>fl/fl</sup> mice as compared to sex-matched littermates at 12 months of age (*Figure 6F*). Calcein double labelling revealed that the mineral apposition rate was significantly reduced in cortical bone from male and female *Lepr-Cre; Itga11*<sup>fl/fl</sup> mice as compared to sex-matched littermate controls at 6 and 12 months of age (*Figure 6G*). Deletion of Integrin α11 from LepR<sup>+</sup> cells thus reduces the rate of cortical bone formation during adulthood, slowly leading to a thinning of cortical bone that became apparent in the femurs at 12 months of age, phenocopying the slow loss of cortical bone in adult *Osteolectin* deficient mice (*Yue et al., 2016*).

## Integrin α11 is required in bone marrow stromal cells to respond to Osteolectin

To test whether Integrin α11 is necessary for the maintenance or the proliferation of skeletal stem/progenitor cells in the bone marrow, we cultured at clonal density enzymatically dissociated femur bone marrow cells from *Lepr-Cre; Itga11*<sup>fl/fl</sup> and sex-matched littermate control mice at 2 and 6 months of age. We observed a slight, but statistically significant, reduction in the frequency of cells that formed CFU-F colonies in *Lepr-Cre; Itga11*<sup>fl/fl</sup> mice at 2 months of age, though no significant difference was apparent at 6 months of age (*Figure 7A*). We observed no significant difference in the number of cells per colony at either age (*Figure 7B*). Integrin α11 is therefore not required for the maintenance of CFU-F in vivo or for their proliferation in vitro.

To test whether integrin α11 regulates the differentiation of bone marrow stromal cells, we cultured CFU-F from *Lepr-Cre; Itga11*<sup>fl/fl</sup> and littermate control mice at clonal density, then replated equal numbers of cells from *Itga11* deficient and control colonies into osteogenic or adipogenic culture conditions (*Figure 7C and D*). We also centrifuged $2 \times 10^5$ CFU-F cells from *Lepr-Cre; Itga11*<sup>fl/fl</sup> and control colonies to form pellets and then cultured them in chondrogenic medium (*Figure 7E*). Consistent with the decreased osteogenesis from *Itga11* deficient mesenchymal cell lines in culture (*Figure 4A*) and the reduced osteogenesis in *Lepr-Cre; Itga11*<sup>fl/fl</sup> mice in vivo (*Figure 5* and *Figure 6*), bone marrow stromal cells from *Lepr-Cre; Itga11*<sup>fl/fl</sup> mice formed significantly less bone in culture as compared to control colonies (*Figure 7C*). This demonstrates that, like Osteolectin, Integrin α11 promotes osteogenesis by bone marrow stromal cells. We did not detect any difference between *Lepr-Cre; Itga11*<sup>fl/fl</sup> and control colonies in adipogenic or chondrogenic differentiation (*Figure 7D and E*). This is also consistent with the Osteolectin deficiency phenotype, which reduced osteogenesis in vitro and in vivo without having any detectable effect on adipogenesis or chondrogenesis (*Yue et al., 2016*).

To test if Integrin α11 is necessary for the osteogenic response of bone marrow stromal cells to Osteolectin, we cultured CFU-F from the bone marrow of 2-month-old *Lepr-Cre; Itga11*<sup>fl/fl</sup> mice and

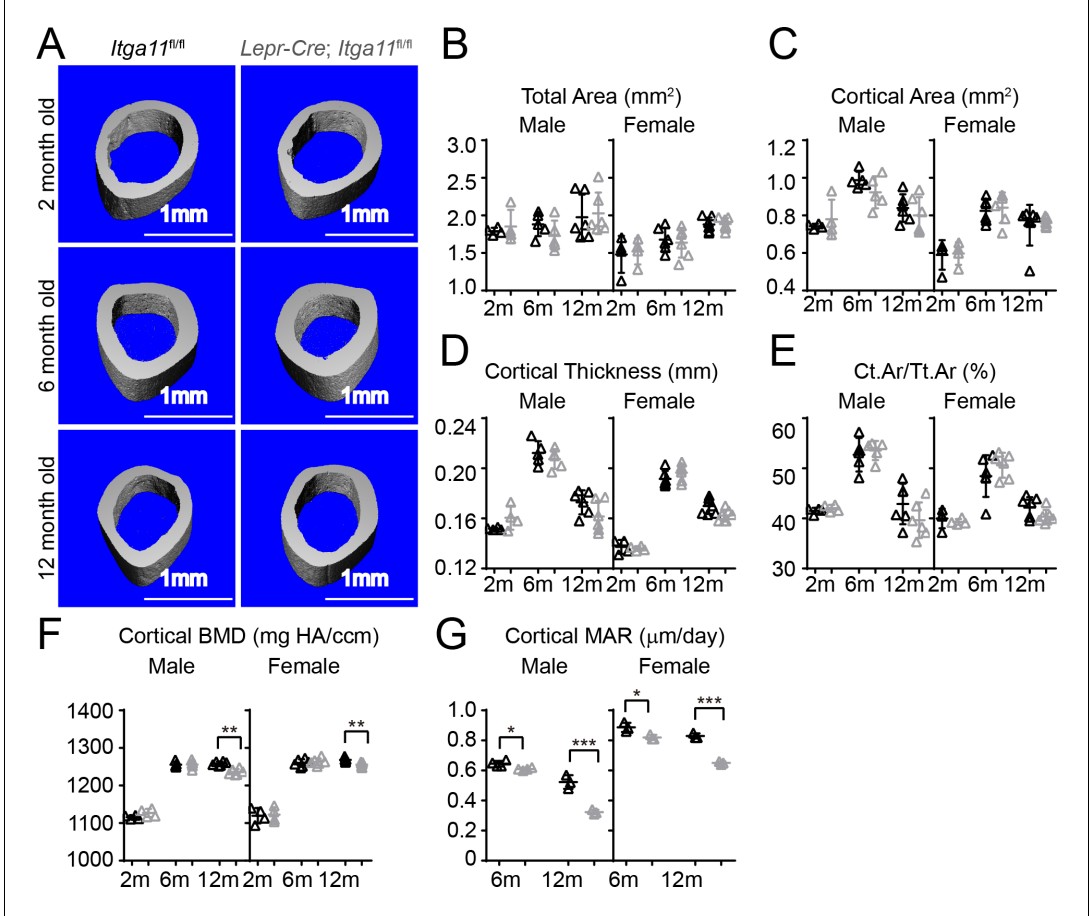

**Figure 6.** Conditional *Itga11* deletion from LepR⁺cells reduces cortical bone formation in adult mice. (A) Representative microCT images of cortical bone in the mid-femur diaphysis of male *Lepr-Cre; Itga11*^fl/fl^ mice and littermate controls at 2, 6, and 12 months of age. (B–F) microCT analysis of the total area (B), cortical area (C), cortical thickness (D), cortical area/total area (E), and cortical bone mineral density (F) in the mid-femur diaphysis of *Lepr-Cre; Itga11*^fl/fl^ mice and sex-matched littermate controls at 2, 6, and 12 months of age (n = 4–7 mice per genotype per sex per time point from at least three independent experiments). (G) Cortical bone mineral apposition rate based on calcein double labelling in the mid-femur diaphysis (n = 3–4 mice per genotype per sex per time point from three independent experiments). All numerical data reflect mean ±standard deviation. The statistical significance of differences was determined with two-way ANOVAs with Sidak's multiple comparisons tests.

DOI: https://doi.org/10.7554/eLife.42274.015

The following source data is available for figure 6:

**Source data 1.** Data for *Figure 6*.

DOI: https://doi.org/10.7554/eLife.42274.016

littermate controls then added osteogenic differentiation medium with or without recombinant mouse Osteolectin. Osteolectin significantly increased osteogenic differentiation by control colonies, but *Lepr-Cre; Itga11*^fl/fl^ colonies underwent significantly less osteogenesis and did not respond to Osteolectin (*Figure 7F and G*). Osteolectin treatment also significantly increased the levels of the Wnt target gene transcripts *Alp*, *Axin2*, *Lef1*, and *Runx2* in cells from control mice, but not *Lepr-Cre; Itga11*^fl/fl^ mice (*Figure 7—figure supplement 1A*). Bone marrow stromal cells thus require Integrin α11 to undergo osteogenesis in response to Osteolectin.

To test if bone marrow stromal cells require Integrin α11 to undergo osteogenesis in response to Osteolectin in vivo, we administered daily subcutaneous injections of recombinant mouse Osteolectin to 2-month-old *Lepr-Cre; Itga11*^fl/fl^ and littermate control mice for 28 days. Osteolectin is functionally important for bone maintenance by 2 months of age given that *Osteolectin* deficient mice exhibit a significant reduction in trabecular bone volume at 2 months of age (*Yue et al., 2016*). Consistent with our prior study (*Yue et al., 2016*), in the distal femur metaphysis of control mice,

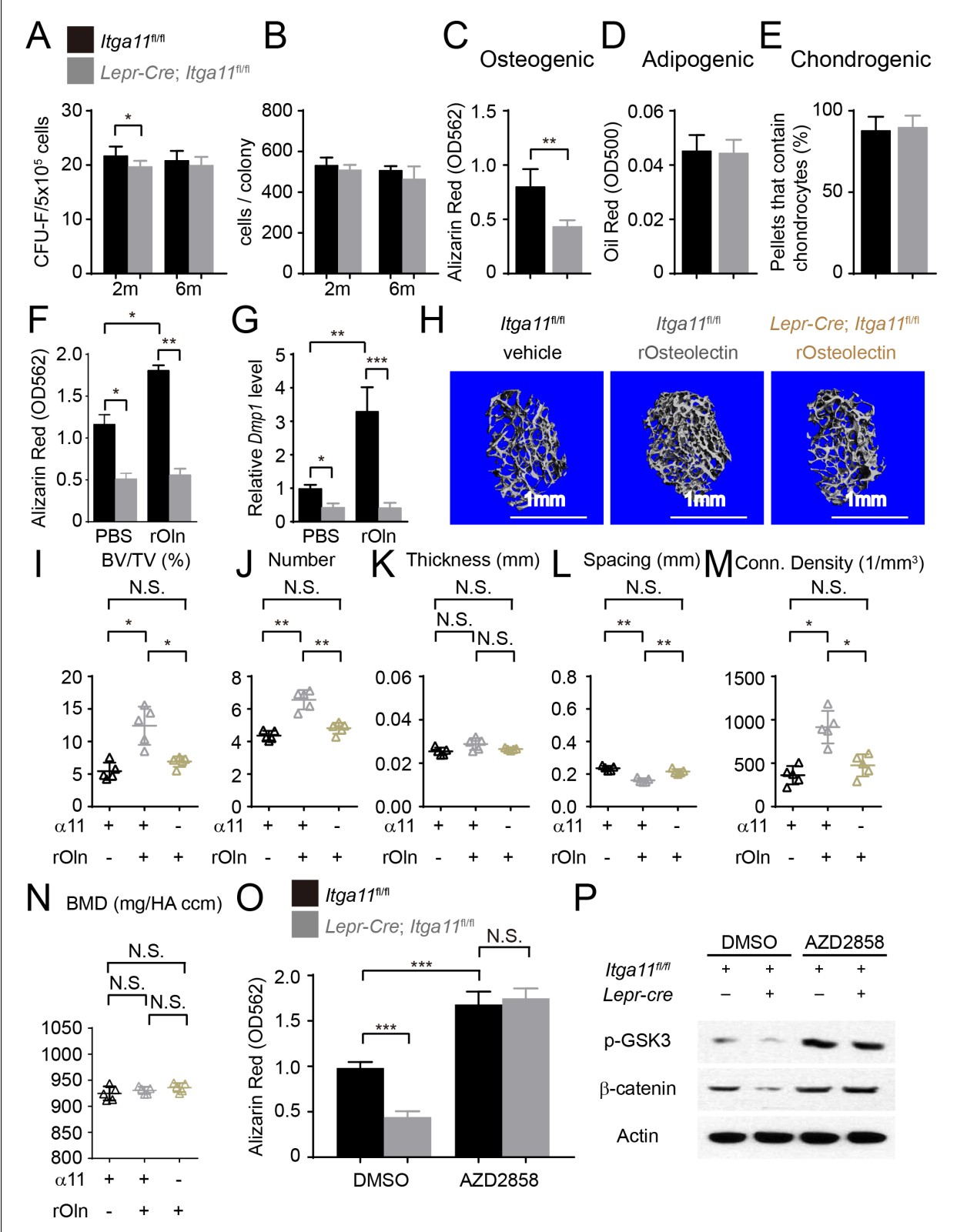

**Figure 7.** Integrin α11 is required by bone marrow stromal cells to undergo osteogenesis in response to Osteolectin. (A) CFU-F frequency and (B) cells per CFU-F colony formed by bone marrow cells from *Lepr-Cre; Itga11*<sup>fl/fl</sup> mice and littermate controls at 2 and 6 months of age (n = 6–8 mice per genotype per time point from at least three independent experiments). (C–E) Osteogenic (C); n = 7 mice per genotype, total, from seven independent experiments), adipogenic (D); n = 6 mice per genotype, total, from six independent experiments), and chondrogenic (E); n = 5 mice per genotype,
*Figure 7 continued on next page*

*Figure 7 continued*

total, from five independent experiments) differentiation of bone marrow stromal cells cultured from the femurs of *Lepr-Cre; Itga11*[fl/fl] mice and sex-matched littermate controls at 2 months of age. (F) Recombinant mouse Osteolectin promoted osteogenic differentiation in culture by femur bone marrow stromal cells from control but not *Lepr-Cre; Itga11*[fl/fl] mice (n = 3 mice per genotype, total, from three independent experiments). (G) qRT-PCR analysis of *Dmp1* transcript levels in cells from panel (F). (H–N) Subcutaneous injection of recombinant mouse Osteolectin daily for 28 days (50 μg/kg body mass/day) significantly increased trabecular bone volume and number in female control mice but not female *Lepr-Cre; Itga11*[fl/fl] mice. (H) Representative microCT images of trabecular bone in the distal femur metaphysis. (I–N) microCT analysis of trabecular bone volume/total volume (I), trabecular number (J), trabecular thickness (K), trabecular spacing (L), connectivity density (M), and bone mineral density (N) in the distal femur metaphysis (n = 5 mice per treatment, total, from three independent experiments). (O) The GSK3 inhibitor, AZD2858, rescued the osteogenic differentiation in culture of bone marrow stromal cells from the femurs of *Lepr-Cre; Itga11*[fl/fl] mice (n = 6 independent experiments in which cells from one mouse of each genotype were cultured in each experiment). (P) Western blotting of cultured cell lysates showed that AZD2858 promoted GSK3 phosphorylation and increased β-catenin levels in bone marrow stromal cells from *Lepr-Cre; Itga11*[fl/fl] mice and littermate controls. All numerical data reflect mean ±standard deviation. The statistical significance of differences was determined with Wilcoxon's test followed by Holm-Sidak multiple comparisons adjustment (A and B), one-way (F), (G,) and O) ANOVAs with Sidak's multiple comparisons tests, by paired t-tests (C–E), or by one-way ANOVAs with Tukey's multiple comparisons tests (I–N).

DOI: https://doi.org/10.7554/eLife.42274.017

The following source data and figure supplements are available for figure 7:

**Source data 1.** Data for *Figure 7*.
DOI: https://doi.org/10.7554/eLife.42274.018
**Figure supplement 1.** Osteolectin promotes Wnt target gene transcription in mouse bone marrow stromal cells in an integrin α11-dependent manner.
DOI: https://doi.org/10.7554/eLife.42274.019
**Figure supplement 1—source data 1.** Data for *Figure 7—figure supplement 1*.
DOI: https://doi.org/10.7554/eLife.42274.020

Osteolectin treatment significantly increased trabecular bone volume (*Figure 7H and I*), trabecular bone number (*Figure 7J*), and trabecular connectivity density (*Figure 7M*), while significantly reducing trabecular spacing (*Figure 7L*). However, Osteolectin treatment had no significant effect on these parameters in *Lepr-Cre; Itga11*[fl/fl] mice (*Figure 7H–N*). Osteolectin treatment also significantly increased the levels of the Wnt target gene transcripts *Alp*, *Lef1*, and *Runx2* in LepR[+] cells isolated from the bone marrow of control mice, but not *Lepr-Cre; Itga11*[fl/fl] mice (*Figure 7—figure supplement 1B*). LepR[+] bone marrow stromal cells and their progeny thus require Integrin α11 to undergo osteogenesis in response to Osteolectin in vivo. Neither Osteolectin administration nor *Itga11* deficiency had any significant effect on cortical bone parameters in this relatively short-term experiment performed in young mice (data not shown).

To test if bone marrow stromal cells from *Lepr-Cre; Itga11*[fl/fl] mice retained the capacity to undergo osteogenesis upon Wnt pathway activation, we cultured these cells from 2-month-old *Lepr-Cre; Itga11*[fl/fl] and littermate control mice and treated half of the cultures with the Wnt pathway agonist, AZD2858. In control cultures, bone marrow stromal cells from *Lepr-Cre; Itga11*[fl/fl] mice underwent significantly less osteogenesis as compared to stromal cells from control mice (*Figure 7O*). Addition of AZD2858 significantly increased osteogenic differentiation from both *Lepr-Cre; Itga11*[fl/fl] and control stromal cells. In cultures treated with DMSO control, *Lepr-Cre; Itga11*[fl/fl] stromal cells had lower levels of phosphorylated GSK3 and β-catenin as compared to control stromal cells (*Figure 7P*). AZD2858 increased the levels of phosphorylated GSK3 and β-catenin in both *Lepr-Cre; Itga11*[fl/fl] and control stromal cells (*Figure 7P*). *Lepr-Cre; Itga11*[fl/fl] stromal cells thus retain the ability to undergo osteogenesis in response to Wnt pathway activation, even though they do not respond to Osteolectin.

## Discussion

Our data demonstrate that integrin α11 is a physiologically important receptor for Osteolectin, mediating its effect on osteogenesis. Integrin α11 is expressed by LepR[+] skeletal stem cells and osteoblasts but shows little expression in non-osteogenic cells (*Figure 1G*). Osteolectin bound selectively to α11β1 integrin, with nanomolar affinity (*Figure 1I and J*), and promoted Wnt pathway activation in bone marrow stromal cells (*Figure 2*). Integrin α11 was required in bone marrow stromal cells for Wnt pathway activation and osteogenesis in response to Osteolectin (*Figure 4A–D*). Blocking Wnt pathway activation in bone marrow stromal cells blocked the osteogenic effect of

Osteolectin (*Figure 3C–D*). Conditional deletion of *Itga11* from LepR⁺ cells phenocopied the effect of *Osteolectin* deficiency (*Yue et al., 2016*): in both cases the mice were grossly normal but exhibited accelerated bone loss during adulthood, particularly in trabecular bone (*Figure 5*). Like *Osteolectin* deficiency (*Yue et al., 2016*), *Itga11* deficiency significantly reduced the rate of bone formation in adult mice (*Figure 5M and O*) without affecting the rate of bone resorption (*Figure 5N*). Bone marrow stromal cells from *Lepr-Cre; Itga11*^fl/fl^ mice differentiated normally to adipocytes and chondrocytes (*Figure 7A–E*) but exhibited reduced osteogenic differentiation and did not respond to Osteolectin in vitro (*Figure 7C, F and G* and *Figure 7—figure supplement 1A*) or in vivo (*Figure 7H–N* and *Figure 7—figure supplement 1B*). Nonetheless, bone marrow stromal cells from *Lepr-Cre; Itga11*^fl/fl^ mice retained the ability to form bone in response to a chemical inhibitor of GSK3, which activates the Wnt pathway (*Figure 7O and P*). We conclude that integrin α11 is required by skeletal stem/progenitor cells to undergo osteogenesis in response to Osteolectin.

Multiple factors promote osteogenesis by activating the Wnt pathway, including Wnts (*Boyden et al., 2002*; *Cui et al., 2011*; *Gong et al., 2001*), BMPs (*Chen et al., 2007*; *Rawadi et al., 2003*), hedgehog proteins (*Mak et al., 2006*), and parathyroid hormone (*Bonnet et al., 2012*; *Kulkarni et al., 2005*; *Wan et al., 2008*). Our data suggest that Osteolectin contributes to Wnt pathway activation in osteogenic stem/progenitor cells along with other factors.

While deficiency for integrin α11 phenocopied the effects of *Osteolectin* deficiency, we do not rule out a potential role for α10 integrin in mediating certain effects of Osteolectin. α10β1 also bound Osteolectin with nanomolar affinity (*Figure 1I*). Integrin α11 is more highly expressed than α10 by LepR⁺ cells (*Figure 1D*); however, integrin α10 is expressed by chondrocytes (*Bengtsson et al., 2005*; *Reinisch et al., 2015*). While we did not observe any cartilage defects in *Osteolectin* deficient mice (*Yue et al., 2016*), Osteolectin may promote the differentiation of hypertrophic chondrocytes into bone in adult mice, such as during fracture healing. Therefore, α10 integrin may mediate the effects of Osteolectin on hypertrophic chondrocytes while integrin α11 may mediate the effects of Osteolectin on skeletal stem/progenitor cells. It also remains possible that osteolectin has other receptors.

Osteolectin may not be the only osteogenic ligand for integrin α11. Collagen is a known ligand for α11β1 integrin (*Popova et al., 2007*). We found that collagen binds to α11β1 with nanomolar affinity (*Figure 1J*) and actives integrin signaling (*Figure 4G and 4H*, but we did not detect any effect of exogenous collagen on β-catenin accumulation (*Figure 4F*) or osteogenic differentiation (*Figure 1K*). This suggests that collagen may bind α11β1 in a way that regulates cell adhesion and migration but not osteogenic differentiation, at least in skeletal stem/progenitor cells. Alternatively, endogenous collagen may bind α11β1 differently than exogenous collagen, potentially promoting osteogenesis. Since *Lepr-Cre; Itga11*^fl/fl^ mice delete *Itga11* in postnatal bone marrow cells that exhibit little contribution to the skeleton prior to two months of age (*Zhou et al., 2014*), it remains untested whether integrin α11 regulates osteogenesis during fetal or early postnatal development. If so, this would raise the possibility of a distinct osteogenic ligand for α11 during development as *Osteolectin* deficient mice do not appear to exhibit defects in skeletal development (*Yue et al., 2016*).

While integrin α11 is not widely expressed by non-osteogenic cells, integrin α11 may have non-osteogenic functions in certain other cell types, or during development, in cells that are not competent to undergo osteogenesis. Integrin α11 is expressed by periodontal ligament fibroblasts and is required for the migration of these cells during ligament development, leading to a failure of tooth eruption in germline *Itga11* deficient mice (*Popova et al., 2007*). This raises the possibility that collagen binding to α11β1 may have biologically distinct consequences in cells that are not competent to form bone.

Given that integrins can function as mechanosensors (*Schwartz, 2010*), our data raise the possibility that integrin α11 mediates the osteogenic response to mechanical loading in bones. Interestingly, it was recently discovered that skeletal stem cells in the developing jaw undergo osteogenesis in response to mechanical forces by activating FAK, suggesting the involvement of integrins in this process (*Ransom et al., 2018*).

In conclusion, we identify integrin α11 as an Osteolectin receptor and a new regulator of osteogenesis and adult skeleton maintenance. The identification of a new ligand/receptor pair that regulates the maintenance of the adult skeleton offers the opportunity to better understand the physiological and pathological mechanisms that influence skeletal homeostasis.

# Materials and methods

## Key resources table

| Reagent type (species) or resource | Designation | Source or reference | Identifiers | Additional information |
|---|---|---|---|---|
| Genetic reagent (*M.musculus*) | *Osteolectin$^{-/-}$* | PMID: 27976999 | | |
| Genetic reagent (*M.musculus*) | *Lepr-Cre* | PMID: 11283374 | | JAX Stock (008320) |
| Antibody | rabbit polyclonal anti-phospho-PI3 Kinase | Cell Signaling | 4228S | (1:1000) |
| Antibody | rabbit polyclonal anti-phospho-Akt (Ser473) | Cell Signaling | 4060S | (1:1000) |
| Antibody | rabbit polyclonal anti-phospho-GSK-3α/β | Cell Signaling | 9331S | (1:1000) |
| Antibody | rabbit polyclonal anti-β-Catenin | Cell Signaling | 8480S | (1:1000) |
| Antibody | rabbit polyclonal anti-anti-β-Actin | Cell Signaling | 8457S | (1:10000) |
| Antibody | rabbit polyclonal anti-GSK-3b | Cell Signaling | 9315S | (1:1000) |
| Antibody | goat polyclonal anti-rabbit IgG, HRP-linked antibody | Cell Signaling | 7074S | (1:5000) |
| Antibody | horse polyclonal anti-mouse IgG, HRP-linked antibody | Cell Signaling | 7076S | (1:5000) |
| Antibody | donkey polyclonal anti-sheep IgG, HRP-linked antibody | R and D Systems | HAF016 | (1:5000) |
| Antibody | rabbit polyclonal anti-phospho-FAK (Y397) | Cell Signaling | 3283S | (1:1000) |
| Antibody | rabbit polyclonal anti-Histone H3 | Cell Signaling | 4499S | (1:5000) |
| Antibody | goat polyclonal anti-mouse Osteolectin | R and D Systems | AF3729 | (1:1000) |
| Antibody | sheep polyclonal anti-human Osteolectin | R and D Systems | AF1904 | (1:1000) |
| Antibody | rabbit polyclonal anti-Adiponectin | Abcam | ab181699 | (1:1000) |
| Antibody | rabbit polyclonal anti-Integrin α11 | Abcam | ab198826 | (1:1000) |
| Antibody | rabbit polyclonal anti-FAK | Cell Signaling | 3285P | (1:1000) |
| Antibody | mouse monoclonal anti-6x-His Tag antibody | Thermo Fisher Scientific | MA1-135 | (1:1000) |
| Antibody | rat polyclonal anti-CD45-APC | Tonbo | 20–0451 | (1:200) |
| Antibody | rat polyclonal anti-Ter119-APC | Tonbo | 20–5921 | (1:200) |
| Antibody | rat polyclonal anti-CD31-APC | Biolegend | 102410 | (1:200) |

*Continued on next page*

*Continued*

| Reagent type (species) or resource | Designation | Source or reference | Identifiers | Additional information |
|---|---|---|---|---|
| Antibody | goat polyclonal anti-Mouse Leptin R, Biotin | R and D Systems | AF497 | (1:200) |
| Antibody | mouse monoclonal anti-Streptavidin PE | Biolegend | 410504 | (1:500) |
| Peptide, recombinant protein | Bovine serum albumin | Sigma-Aldrich | A3156 | |
| Peptide, recombinant protein | recombinant human pro-Collagen I α1 | R and D Systems | 6220 CL | |
| Peptide, recombinant protein | recombinant human osteolectin | PMID: 27976999 | | |
| Peptide, recombinant protein | recombinant mouse osteolectin | PMID: 27976999 | | |
| Peptide, recombinant protein | recombiant Integrin α11β1 protein | R and D Systems | 6357-AB | |
| Peptide, recombinant protein | recombiant Integrin α10β1 protein | R and D Systems | 5895-AB | |
| Peptide, recombinant protein | recombiant Integrin αVβ3 protein | R and D Systems | 3050-AV | |
| Peptide, recombinant protein | recombiant Integrin αVβ1 Protein | R and D Systems | 6579-AV | |
| Peptide, recombinant protein | recombiant Integrin α4β1 protein | R and D Systems | 5668-A4 | |
| Peptide, recombinant protein | recombiant Integrin α9β1 protein | R and D Systems | 5438-A9 | |
| Peptide, recombinant protein | recombiant Integrin αIIbβ3 Protein | R and D Systems | 7148-A2 | |
| Peptide, recombinant protein | recombiant Integrin αMβ2 Protein | R and D Systems | 4047-AM | |
| Chemical compound, drug | DAPI | Life Technologies | D1306 | |
| Chemical compound, drug | TRIzol LS Reagent | Invitrogen | 10296028 | |
| Chemical cmpound, drug | Collagenase, Type 1 | Worthington | LS004196 | |
| Chemical compound, drug | Dispase II | Roche Diagnostic | D4693 | |
| Chemical compound, drug | DNase I | Sigma-Aldrich | 10 104 159 001 | |
| Chemical compound, drug | IWR-1-endo | Sigma-Aldrich | 681669 | |
| Chemical compound, drug | AZD2858 | Selleck | S7253 | |
| Chemical compound, drug | Y-27632 Rock inhibitor | Selleck | S1049 | |
| Chemical compound, drug | 4% paraformaldehyde in PBS | Thermo Fisher Scientific | J19943-K2 | |

*Continued on next page*

*Continued*

| Reagent type (species) or resource | Designation | Source or reference | Identifiers | Additional information |
|---|---|---|---|---|
| Cell lines (*M.musculus*) | MC3T3-E1, subclone 4 | ATCC | CRL-2593 | |
| Cell lines (*H.sapiens*) | hBMSC#1 | ATCC | PCS-500–012 | |
| Cell lines (*H.sapiens*) | hBMSC#2 | Lonza | PT-2501 | |
| Commercial assay or kit | Cell Culture Contamination Detection Kit | Molecular Probes | C-7028 | |
| Commercial assay or kit | TMB stop solution | KPL | 95059–198 | |
| Commercial assay or kit | SureBlue TMB Microwell Peroxidase Substrate | KPL | 95059–282 | |
| Commercial assay or kit | HBSS, without Calcium and Magnesium | Corning | 21022CV | |
| Commercial assay or kit | HBSS, with Calcium and Magnesium | Corning | 21023CV | |
| Commercial assay or kit | CleanCap Cas9 mRNA | TriLink | L-7206 | |
| Commercial assay or kit | MEGAshortscript T7 Transcription Kit | Ambion | AM1354 | |
| Commercial assay or kit | MEGAclear Transcription Clean-Up Kit | Ambion | AM1908 | |
| Commercial assay or kit | EDTA-free Protease Inhibitor Cocktail | Sigma-Aldrich | 11 836 170 001 | |
| Commercial assay or kit | ECL Western Blotting Substrate | Pierce | 32106 | |
| Commercial assay or kit | SuperScript III Reverse Transcriptase | Invitrogen | 18080044 | |
| Commercial assay or kit | StemPro Osteogenesis Differentiation Kit | Gibco | A1007201 | |
| Commercial assay or kit | StemPro Adipogenesis Differentiation Kit | Gibco | A1007001 | |
| Commercial assay or kit | StemPro Chondrogenesis Differentiation Kit | Gibco | A1007101 | |
| Commercial assay or kit | MicroVue DPD ELISA Kit | Quidel | 8007 | |
| Commercial assay or kit | MicroVue Creatinine Assay Kit | Quidel | 8009 | |
| Commercial assay or kit | Rat/Mouse PINP ELISA kit | Immunodiagnostic Systems | AC-33F1 | |
| Commercial assay or kit | Nuclear/Cytosol Fractionation Kit | Biovision | K266 | |
| Other | Fetal bovine serum | Sigma-Aldrich | F2442 | |
| Other | Penicillin Streptomycin 100x solution | HyClone | SV3001 | |
| Other | DMEM, low glucose | Gibco | 11885084 | |
| Other | DMEM, high glucose | Gibco | D5671 | |

## Mice and cell lines

*Lepr-cre* mice were described previously (*DeFalco et al., 2001*) and obtained from the Jackson Laboratory (Stock No: 008320). *Lepr-cre* mice were backcrossed at least eight times onto a C57BL/Ka

background. To generate *Itga11*<sup>fl/fl</sup> mice, CleanCap Cas9 mRNA (TriLink) and sgRNAs (transcribed using MEGAshortscript Kit (Ambion), purified using the MEGAclear Kit (Ambion)), and recombineering plasmids were microinjected into C57BL/Ka zygotes. Chimeric mice were genotyped by restriction fragment length poly-morphism (RFLP) analysis and confirmed by Southern blotting and sequencing of the targeted allele. Founders were mated with C57BL/Ka mice to obtain germline transmission then backcrossed with wild-type C57BL/Ka mice for at least three generations prior to analysis. This study was performed in accordance with the recommendations in the Guide for the Care and Use of Laboratory Animals of the National Institutes of Health. All procedures were approved by the UTSW Institutional Animal Care and Use Committee (protocol number 2016–101334 G).

Cell lines used in this study included mouse preosteoblast MC3T3-E1 cells (Subclone 4, ATCC CRL-2593), human bone marrow stromal cells from ATCC (PCS-500–012; referred to as hBMSC#1), and human bone marrow stromal cells from Lonza (PT-2501, hBMSC#2). The identity of MC3T3-E1 cells has been authenticated by ATCC, based on their expression of osteoblast marker genes including *Bsp*, *Ocn*, *Pth* and *Pthrp*. The identity of hBMSC#1 cells has been authenticated by ATCC using cell surface markes for these cells, including CD105, CD73, and CD44. The identity of hBMSC#2 cells has been authenticated by Lonza using cell surface markes for these cells, including CD105, CD166, CD73, and CD44. We found no contamination of these cells from yeast, fungi, gram-positive or gram-negative bateria using the Cell Culture Contamination Detection Kit (Molecular Probes). MC3T3-E1 cells were cultured in Alpha Minimum Essential Medium with ribonucleosides, deoxyribonucleosides, 2 mM L-glutamine and 1 mM sodium pyruvate, but without ascorbic acid (GIBCO, A1049001), and supplemented with 10% fetal bovine serum (Sigma, F2442) and penicillin-streptomycin (HyClone). MC3T3-E1 cells were used for experiments before passage 20. hBMSC cells were cultured in low glucose DMEM (Gibco) supplemented with 20% fetal bovine serum (Sigma, F2442) and penicillin-streptomycin (HyClone), and were used for experiments before passage 16.

## Recombinant protein production and use

As described previously (*Yue et al., 2016*), mouse and human Osteolectin cDNA were cloned into pcDNA3 vector (Invitrogen) containing a C-terminal 1XFlag-tag and transfected into HEK293 cells with *Akashi et al. (2000)* (Invitrogen). Stably expressing cell lines were selected using 1 mg/ml G418 (Sigma) then cultured in DMEM plus 10% FBS (Sigma), and 1% penicillin/streptomycin (Invitrogen). Culture medium was collected every two days, centrifuged to eliminate cellular debris, and stored with 1 mM phenylmethylsulfonyl fluoride (Sigma) at 4°C to inhibit protease activity. One liter of culture medium was filtered through a 0.2 µm membrane (Nalgene) to eliminate cellular debris before being loaded onto a chromatography column containing 2 ml Anti-FLAG M2 Affinity Gel (Sigma), with a flow rate of 1 ml/min. The column was sequentially washed using 20 ml of high salt buffer (20 mM Tris-HCl, 300 mM KCl, 10% Glycerol, 0.2 mM EDTA) followed by 20 ml of low salt buffer (20 mM Tris-HCl, 150 mM KCl, 10% Glycerol, 0.2 mM EDTA) and finally 20 ml of PBS. The FLAG-tagged Osteolectin was then eluted from the column using 10 ml 3X FLAG peptide (100 mg/ml) in PBS. Eluted protein was concentrated using Amicon Ultra-15 Centrifugal Filter Units (Ultracel-10K, Millipore), then quantitated by SDS-PAGE and colloidal blue staining (Invitrogen) and stored at −80°C.

Recobinant human Pro-Collagen 1α was purchased from R and D Systems, and we removed the His tag using TEV protease (Sigma). After cleavage, we purified the untagged Pro-Collagen 1α using Ni-NTA agarose columns (Thermo Fisher Scientific) to separate it from the cleaved His tag, the His-tagged Pro-Collagen 1α, and the His-tagged TEV protease.

To add recombinant proteins in culture, recombinant human or mouse Osteolectin or Pro-Collagen 1α was added to osteogenic differentiation medium (described below). Unless otherwise specified, we used 30 ng/ml recombinant Osteolectin for in vitro assays. For in vivo use, recombinant mouse Osteolectin (50 µg/kg of body mass) was subcutaneously injected daily into 2-month-old female *Lepr-Cre; Itga11*<sup>fl/fl</sup> or littermate *Itga11*<sup>fl/fl</sup> control mice for 28 days. Mice receiving control injections received an equal volume of PBS.

## Western blots and co-immunoprecipitation

Cells were cultured until confluent, then transferred into osteogenic differentiation medium with or without Osteolectin or small molecule inhibitors of Wnt pathway components. Prior to extracting

proteins, cells were washed with PBS and then lysis buffer was added containing 50 mM Tris-HCl, 150 mM NaCl, 1% NP-40, 0.5% sodium deoxycholate, 0.1% SDS, 1 mM sodium vanadate, 0.5 mM sodium fluoride, and cOmplete Mini EDTA-free Protease Inhibitor Cocktail (Sigma). The cells were scraped off the plate in the lysis buffer, transferred to an Eppendorf tube on ice, incubated for 20 min with occasional vortexing, then centrifuged at 17,000xg for 10 min at 4°C to clear cellular debris. The cell lysates were Western blotted with the indicated antibodies and immunoreactive bands were detected using ECL reagent (Pierce). For some experiments, a Nuclear/Cytosol Fractionation Kit (Biovision) was used to separate the nuclear and cytosolic/membrane fractions of cell lysates. Antibodies used in this study include anti-Phospho-PI3 Kinase p85(Tyr458)/p55(Tyr199), anti-Phospho-Akt (Ser473), anti-Phospho-GSK-3α/β (Ser21/9), anti-Phospho-FAK (Y397), anti-β-Catenin, anti-β-Actin, anti-GSK-3β (27C10), anti-Histone H3, and anti-rabbit IgG, HRP-linked antibody from Cell Signaling, anti-mouse Osteolectin (AF3729) and anti-human Osteolectin (AF1904) antibodies from R and D Systems, anti-Adiponectin (ab181699) and anti-Integrin α11 antibody (ab198826) from Abcam.

For co-immunoprecipitation experiments, human Osteolectin cDNA was cloned into pcDNA3 vector (Invitrogen) containing a C-terminal 1XFlag-tag, then transfected into MC3T3-E1 cells with *Morrison et al., 2000* (Invitrogen). After 48 hr, cells were solubilized in lysis buffer and cellular debris was cleared by centrifugation as described above, then lysates were immunoprecipitated with anti-FLAG M2 Affinity Gel (Sigma). After incubation of lysates with M2 Affinity Gel for 2 hr at 4°C, the gel was centrifuged and washed six times with lysis buffer. Immunoprecipitates were analyzed by western blotting.

## qRT-PCR

For quantitative reverse transcription PCR (qRT-PCR), cells were lysed using TRIzol LS (Invitrogen). RNA was extracted and reverse transcribed into cDNA using SuperScript III (Invitrogen). qRT-PCR was performed using a Roche LightCycler 480. The primers used for qRT-PCR analysis of mouse RNA include: *Osteolectin*: 5'-AGG TCC TGG GAG GGA GTG-3' and 5'-GGG CCT CCT GGA GAT TCT T-3'; *Actb*: 5'-GCT CTT TTC CAG CCT TCC TT-3' and 5'-CTT CTG CAT CCT GTC AGC AA-3'; *Lef1*: 5'-TGT TTA TCC CAT CAC GGG TGG-3' and 5'-CAT GGA AGT GTC GCC TGA CAG-3'; *Runx2*: 5'-TTA CCT ACA CCC CGC CAG TC-3' and 5'-TGC TGG TCT GGA AGG GTC C-3'; *Axin2*: 5'-GAG TAG CGC CGT GTT AGT GAC T-3' and 5'-CCA GGA AAG TCC GGA AGA GGT ATG-3'; *Alp*: 5'-CCA ACT CTT TTG TGC CAG AGA-3' and 5'-GGC TAC ATT GGT GTT GAG CTT TT-3', *Rankl*: 5'-CAG CAT CGC TCT GTT CCT GTA-3' and 5'-CTG CGT TTT CAT GGA GTC TCA-3', *Itga11*: 5'-TGC CCC AAT GGA AAC CAA TG-3' and 5'-CAC TCG TGC GAC CAG AGA G-3', *Dmp1*: 5'-TGG GAG CCA GAG AGG GTA G-3' and 5'-TTG TGG TAT CTG GCA ACT GG-3', *Ctnnb1*: 5'-CAT CTA CAC AGT TTG ATG CTG CT-3' and 5'-GCA GTT TTG TCA GTT CAG GGA-3'. The primers used for qRT-PCR analysis of human RNA include: *Osteolectin*: 5'-ACA TCG TCA CTT ACA TCC TGG GC-3' and 5'-CAC GCG GGT GTC CAA CG-3'; *Actb*: 5'-ATT GGC AAT GAG CGG TTC-3' and 5'-CGT GGA TGC CAC AGG ACT-3'; *Lef1*: 5'-TGC CAA ATA TGA ATA ACG ACC CA-3' and 5'-GAG AAA AGT GCT CGT CAC TGT-3'; *Runx2*: 5'-GAA CCC AGA AGG CAC AGA CA-3' and 5'-GGC TCA GGT AGG AGG GGT AA-3'; *Axin2*: 5'- CAA CAC CAG GCG GAA CGA A-3' and 5'- GCC CAA TAA GGA GTG TAA GGA CT-3'; *Alp*: 5'-GTG AAC CGC AAC TGG TAC TC-3' and 5'-GAG CTG CGT AGC GAT GTC C-3', *Dmp1*: 5'-CTC CGA GTT GGA CGA TGA GG-3' and 5'-TCA TGC CTG CAC TGT TCA TTC-3'.

## PCR genotyping

To genotype *Itga11* floxed mice the following primers were used: 5'- AATTCAGTGCCGATCC TCCAGTGTC-3', 5'-CCCTTGCTTCCTTCTGCTGTCACTT-3' (*Itga11*[fl] allele: 370 bp; *Itga11*[+] allele: 280 bp).

## Integrin binding assay

Integrin binding assays were performed as described (*Nishiuchi et al., 2006*). Microtiter plates were coated with 10 nM recombinant human Osteolectin, recombinant human Pro-Collagen 1α, or Bovine Serum Albumin (BSA, Sigma A3156) overnight at 4°C, and then blocked with 10 mg/ml BSA. 6xHis tagged recombinant human integrin heterodimers were purchased from R and D Systems. The

plates were incubated with integrins in TBS buffer (50 mM Tris-Cl, pH 7.5 150 mM NaCl) with 1 mM $MnCl_2$, then washed with TBS containing 1 mM $MnCl_2$, 0.1% BSA, and 0.02% Tween 20, followed by quantification of bound integrins by an enzyme-linked immunosorbent assay using an anti-His tag monoclonal antibody (Thermo Fisher Scientific, clone 4E3D10H2/E3) followed by a horseradish per-oxidase-conjugated anti-mouse secondary antibody. After washing, bound HRP was detected using SureBlue TMB Microwell Peroxidase Substrate (KPL) and the reaction was stopped with TMB stop solution (KPL). The optical density was measured at 450 nm.

## MicroCT analysis

MicroCT analysis was performed using the same settings as previously described (*Yue et al., 2016*). Based on previously described methods (*Bouxsein et al., 2010*), mouse femurs were dissected, fixed overnight in 4% paraformaldehyde (Thermo Fisher Scientific) and stored in 70% ethanol at 4°C. Femurs and lumbar vertebrae were scanned at an isotropic voxel size of 3.5 μm and 7 μm, respectively, with peak tube voltage of 55 kV and current of 0.145 mA (μCT 35; Scanco). A three-dimensional Gaussian filter (s = 0.8) with a limited, finite filter support of one was used to suppress noise in the images, and a threshold of 263–1000 was used to segment mineralized bone from air and soft tissues. Trabecular bone parameters were measured in the distal metaphysis of the femurs. The region of interest was selected from below the distal growth plate where the epiphyseal cap struc-ture completely disappeared and continued for 100 slices toward the proximal end of the femur. Contours were drawn manually a few voxels away from the endocortical surface to define trabecular bones in the metaphysis. Cortical bone parameters were measured by analyzing 100 slices in mid-diaphysis femurs.

## Statistical analysis

Numbers of experiments noted in figure legends reflect independent experiments performed on dif-ferent days. Mice were allocated to experiments randomly and samples processed in an arbitrary order, but formal randomization techniques were not used. Prior to analyzing the statistical signifi-cance of differences among treatments we tested whether data were normally distributed and whether variance was similar among treatments. To test for normality, we performed the Shapiro–Wilk tests. To test whether variability significantly differed among treatments we performed $F$-tests (for experiments with two treatments) or Levene's median tests (for experiments with more than two treatments). When the data significantly deviated from normality ($p < 0.01$) or variability significantly differed among treatments ($p < 0.05$), we $log_2$-transformed the data and tested again for normality and variability. If the transformed data no longer significantly deviated from normality and equal var-iability, we then performed parametric tests on the transformed data. If the transformed data still significantly deviated from normality or equal variability, we performed non-parametric tests on the non-transformed data. Data from the same cell culture experiments were always paired for statistical analysis. Mouse littermates were paired for statistical analysis.

To assess the statistical significance of a difference between two treatments, we used paired two-tailed Student's $t$-tests (when a parametric test was appropriate) or Wilcoxon's tests (when a non-parametric test was appropriate). To assess the statistical significance of differences between more than two treatments, we used one-way or two-way repeated measures ANOVAs (when a parametric test was appropriate) followed by post-hoc tests including Dunnett's, Sidak's, and Tukey's tests depending on the experimental settings and planned comparisons, or multiple Wilcoxon's tests fol-lowed by Holm-Sidak's method for multiple comparisons adjustment (when a non-parametric test was appropriate). Relative mRNA levels were always log2-transformed before any statistical tests were performed. All statistical analyses were performed with Graphpad Prism 7.02. All data repre-sent mean ±standard deviation (*$p < 0.05$, **$p < 0.01$, ***$p < 0.001$).

## Bone marrow digestion and CFU-F assay

As previously described (*Yue et al., 2016*), mouse femurs and tibias were cut at both ends to flush out intact marrow plugs. Both the flushed plugs and crushed bone metaphyses were subjected to two rounds of enzymatic digestion in prewarmed digestion buffer containing 3 mg/ml type I collage-nase (Worthington), 4 mg/ml dispase (Roche Diagnostic) and 1 U/ml DNase I (Sigma) in HBSS with calcium and magnesium, at 37°C for 15 min each round. During each round of digestion, the

suspension was vortexed six times for 10 s each time at speed level three using a Vortex-Genie two to promote more complete dissociation. Dissociated cells were transferred into a tube with staining medium (HBSS without calcium and magnesium +2% fetal bovine serum) and 2 mM EDTA to stop the digestion. Cells were then centrifuged, resuspended in staining medium, and passed through a 90 µm nylon mesh to filter undigested plugs or bone.

To form CFU-F colonies, freshly dissociated bone marrow cell suspensions were plated at clonal density in 6-well plates ($5 \times 10^5$ cells/well) or 10 cm plates ($5 \times 10^6$ cells/dish) with DMEM (Gibco) plus 20% fetal bovine serum (Sigma F2442), 10 mM ROCK inhibitor (Y-27632, Selleck), and 1% penicillin/streptomycin (Invitrogen) at 37°C in gas-tight chambers (Billups-Rothenberg) with 1% $O_2$ and 6% $CO_2$ (with balance Nitrogen) to maintain a low oxygen environment that promoted survival and proliferation (*Morrison et al., 2000*). The CFU-F culture dish was rinsed with HBBS without calcium and magnesium and replenished with freshly made medium on the second day after plating to wash out contaminating macrophages. Cultures were then maintained in a gas-tight chamber that was flushed daily for 1 min with a custom low oxygen gas mixture (1% $O_2$, 6% $CO_2$, balance Nitrogen). The culture medium was changed every 4 days. To count CFU-F colonies, the cultures were stained with 0.1% Toluidine blue in 4% formalin solution eight days after plating.

## Differentiation assays in culture

The osteogenic potential of primary CFU-F cells, human bone marrow stromal cells, and MC3T3-E1 cells was assessed by plating the cells into 48- well plates (25,000 cells/cm$^2$). On the second day after plating, the culture medium was replaced with osteogenic differentiation medium (StemPro Osteogenesis Differentiation kit, Gibco). Cells were maintained in the differentiation medium, with medium change every other day for 14 days for primary CFU-F cells and MC3T3-E1 cells before differentiation was assessed. For human marrow stromal cells, the culture medium was changed every 3 days for 21 days. Osteoblastic differentiation was detected by staining with Alizarin red S (Sigma). To quantitate Alizarin red staining, the stained cells were rinsed with PBS, and extracted with 10% (w/v) cetylpyridinium chloride in 10 mM sodium phosphate, pH 7.0 for 10 min at room temperature. Alizarin red in the extract was quantitated by optical density measurement at 562 nm.

The adipogenic potential of CFU-F cells was assessed by plating them into 48-well plates (25,000 cells/cm$^2$). On the second day after plating, the culture medium was replaced with adipogenic differentiation medium (StemPro Adipogenesis Differentiation Kit, Gibco) and the cultures were allowed to differentiate for 14 days, with culture medium changed every 3 days. Adipocyte differentiation was detected by staining with Oil red O (Sigma). To quantitate the amount of Oil red O staining, cells were rinsed with PBS, and extracted with 100% isopropanol for 10 min at room temperature. Oil red O in the extract was quantitated by optical density measurement at 500 nm.

The chondrogenic potential of CFU-F cells was assessed by centrifuging $2 \times 10^5$ cells to form cell pellets, which were then cultured in chondrogenic medium (StemPro chondrogenesis differentiation kit; Gibco) for 21 days. The culture medium was changed every 3 days. Chondrocyte formation within the cell pellets was assessed by cryosectioning and Toluidine blue staining as described (*Robey et al., 2014*).

## Calcein double labeling and histomorphometry analysis

As previously described (*Egan et al., 2012*), mice were injected intraperitoneally with 10 mg/kg body weight of calcein, dissolved in 0.15 M NaCl plus 2% NaHCO$_3$ in water, at day 0 and day 7. Mice were sacrificed on day 9. Mouse tibias were fixed overnight in 4% paraformaldehyde at 4°C, dehydrated in 30% sucrose at 4°C for two days and sectioned without decalcification (7 µm sections). Mineral apposition rates were determined as previously described (*Egan et al., 2012*). The surface used to quantify the trabecular bone mineral apposition rate was 100 µm distal to the growth plate and 50 µm in from the endosteal cortical bone of the femur. The surface used to quantify cortical bone mineral apposition rate was the medial endosteal cortical bone surface of the femur.

## ELISA assays

The bone resorption rate was determined by measuring urinary levels of deoxypyridinoline (DPD) using a MicroVue DPD ELISA Kit (Quidel). The DPD values were normalized to urinary creatinine levels using the MicroVue Creatinine Assay Kit (Quidel). The bone formation rate was determined by

measuring serum levels of Procollagen type 1 N-terminal Propeptide (P1NP) using the Rat/Mouse P1NP ELISA kit (Immunodiagnostic Systems). The ELISA assay for Osteolectin was described previously (*Yue et al., 2016*).

## Acknowledgements

SJM is a Howard Hughes Medical Institute (HHMI) Investigator, the Mary McDermott Cook Chair in Pediatric Genetics, the Kathryn and Gene Bishop Distinguished Chair in Pediatric Research, the director of the Hamon Laboratory for Stem Cells and Cancer, and a Cancer Prevention and Research Institute of Texas Scholar. BS was supported by a Ruth L. Kirschstein National Research Service Award (NRSA) Postdoctoral Fellowship (F32) from the National Heart, Lung, and Blood Institute (1F32HL139016-01). AT was supported by the Leopoldina Fellowship Program (LPDS 2016–16) from the German National Academy of Sciences. We thank Nicolas Loof and the Moody Foundation Flow Cytometry Facility, Albert Gross for mouse colony management, Yu Zhang and Hao Zhu for intracytoplasmic sperm injection. This work was supported by the National Institute on Aging (R37 AG02494514).

## Additional information

### Competing interests

Sean J Morrison: Senior editor, *eLife*. Rui Yue: Was an inventor on a pending patent application claiming the use of Osteolectin to promote bone formation. The patent application has not been licensed, so has no existing financial interest in it. WO patent application number: WO2016172026A1. The other authors declare that no competing interests exist.

### Funding

| Funder | Grant reference number | Author |
|---|---|---|
| National Heart, Lung, and Blood Institute | 1F32HL139016-01 | Bo Shen |
| German National Academy of Sciences | LPDS 2016–16 | Alpaslan Tasdogan |
| Howard Hughes Medical Institute | | Sean J Morrison |
| National Institutes of Health | AG02494514 | Sean J Morrison |

The funders had no role in study design, data collection and interpretation, or the decision to submit the work for publication.

### Author contributions

Bo Shen, Conceptualization, Formal analysis, Investigation, Writing—original draft, Writing—review and editing; Kristy Vardy, Payton Hughes, Alpaslan Tasdogan, Rui Yue, Genevieve M Crane, Investigation; Zhiyu Zhao, Formal analysis, Writing—review and editing; Sean J Morrison, Conceptualization, Resources, Formal analysis, Supervision, Funding acquisition, Writing—original draft, Project administration, Writing—review and editing

### Author ORCIDs

Bo Shen http://orcid.org/0000-0002-5237-6144
Zhiyu Zhao http://orcid.org/0000-0001-6308-6997
Genevieve M Crane https://orcid.org/0000-0001-9274-0214
Sean J Morrison http://orcid.org/0000-0003-1587-8329

### Ethics

Animal experimentation: This study was performed in accordance with the recommendations in the Guide for the Care and Use of Laboratory Animals of the National Institutes of Health. All

procedures were approved by the UT Southwestern Institutional Animal Care and Use Committee (protocol number 2016-101334-G).

## Decision letter and Author response

Decision letter https://doi.org/10.7554/eLife.42274.025
Author response https://doi.org/10.7554/eLife.42274.026

## Additional files

### Supplementary files

• Supplementary file 1. Markers used for the flow cytometric isolation of bone marrow cell populations. Stem and progenitor cell populations were isolated from mouse bone marrow by flow cytometry using the listed markers. Lineage (Lin) markers used to isolate Lineage negative cell populations were CD2, CD3, CD5, CD8, Ter119, Gr-1, and B220.
DOI: https://doi.org/10.7554/eLife.42274.021

• Transparent reporting form
DOI: https://doi.org/10.7554/eLife.42274.022

### Data availability

Source data files have been provided for all figures.

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
