## [Decision Letter]

Thank you for submitting your article "Integrin α11 is an Osteolectin receptor and is required for the maintenance of adult skeletal bone mass" for consideration by *eLife*. Your article has been reviewed by three peer reviewers, and the evaluation has been overseen by Clifford Rosen as the Reviewing Editor and Didier Stainier as the Senior Editor. The following individual involved in review of your submission has agreed to reveal his identity: John Lees-Shepard (Reviewer #2).

As such, the reviewers have discussed the reviews with one another and the Reviewing Editor has drafted this decision to help you prepare a revised submission.

Overall there was consensus that the manuscript could break new ground in our understanding of adult stem cell fate and the mechanism of osteolectin in directing lineage through the *Itga11* receptor. In general the experiments are illustrative and the results are relatively supportive of the conclusions. However there were several issues in your revision that need to be resolved before a final decision can be rendered. A major concern is the relatively mild skeletal phenotype with the *LepRCre Itga11* at 12 months, the lack of studies with inhibitors to define the specificity of the integrin receptor, absence of any integrin signaling studies, concerns about confounding by collagen interactions and details about the recombinant osteolectin experiments in which there were inconsistent Wnt signaling. These issues can certainly be addressed in a revision with a greater 'n' for the in vitro studies, more phenotyping of the conditional model, and experiments related to the role of collagen in modulating integrin signaling, as well as competitive inhibitors for integrin signaling. A summary of the three reviewers comments are noted below:

*Reviewer #1:*

The authors continue their previous studies on Lepr+ stromal cells and the role of osteolectin in bone formation. Here they explored the receptor for osteolectin and provided evidence that integrin α11 is an osteolectin receptor mediating bone formation. A main concern has to do with the generally mild bone effect of *Itga11* deletion by a constitutively active Cre (*Lepr-Cre*) that acts in not only osteoprogenitors but also many other cell types including adipocytes. The fact that the bone phenotype does not manifest until much later in life (12 months) makes it very difficult to discern any direct effect. This is an important point as it has become clear that bone mass is regulated by many other tissues outside the skeleton. Furthermore, one cannot discount the potential effect of the integrin deletion on collagen I binding in vivo which may in turn be responsible for the bone phenotype. Other comments are as follows.

"To refine potential candidate receptors for Osteolectin we more closelyexamined the expression of these genes in LepR+ cells, which are osteogenic and should express the receptor, and endothelial cells, which are not osteogenic and should not express the receptor. *Itga1, Itga6*, and *ItgaV* were strongly expressed by both LepR+ cells and endothelial cells, suggesting they did not encode the receptor (Figure 1F). *Itga11* was expressed exclusively by LepR+ cells, not by endothelial cells or other bone marrow cells (Figure 1F)." The reasoning is not sound as there is no evidence that osteolectin/receptor signaling alone is sufficient to induce osteogenesis. Therefore being expressed in nonosteogenic cells should not be an exclusion criterion for the potential receptor.

Details about production and application of recombinant osteolectin and pro-Collagen I used in Figure 1K are missing. Were collagen I used to coat the culture plates, which may be necessary to engage the integrins? Some testing of integrin activation is necessary as a positive control in order for any conclusions to be drawn from a negative result here. Same applies to Figure 4D and 4E. Also in Materials and methods, details about in vivo application of recombinant osteolectin (dose, age of mice etc.) is missing.

Figure 2B, Alizarin red staining alone is not sufficient to support claims about OB differentiation especially since the authors deleted an extracellular matrix protein that may directly affect nucleation of minerals. Same criticism applies to Figure 4A, 7C, 7F.

Figure 2B, the authors observed no increase in β-catenin within 1 hr of osteolectin treatment but only after 24 hrs. This data argues against a direct effect of osteolection on β-catenin signaling. Similarly, the role of β-catenin in osteolectin-induced osteogenesis is also likely indirect. Considering the known function of β-catenin in osteoblast differentiation it is expected that β-catenin depletion would suppress osteogenesis by any upstream osteogenic signals.

Figure 6. The authors show no change in cortical total area (6B) or cortical area (6C) but a significant decrease in cortical thickness (6D) and Ct Ar/Tt Ar (6E). How do they explain that? Figure 6G, what is the surface used for MAR measurements? In general, how did the authors control for effects of genetic background variations on the parameters?

On multiple occasions, the authors seem to imply MC3T3-E1 cells as bone marrow stromal cells, but they are cell lines derived from the newborn mouse calvaria, and generally considered to be more advanced along the OB lineage than the BM stromal cells. These should be corrected. " MC3T3-E1 cells, hBMSC#1 cells, and hBMSC#2 cells secrete Osteolectin into the culture medium (Figure 2A), consistent with our observation that Osteolectin is synthesized by a subset of LepR+ bone marrow stromal cells (Yue et al., 2016)." "Addition of recombinant Osteolectin to culture significantly promoted osteogenesis by parental, but not *Itga11* deficient, MC3T3-E1,hBMSC#1, and hBMSC#2 cells (Figure 4A). Integrin α11 is therefore required by mouse and human bone marrow stromal cells to undergo osteogenesis in response to Osteolectin." "Integrin α11 is therefore required by mouse and human bone marrow stromal cells to activate Wnt pathway signaling in response to Osteolectin."

*Reviewer #2:*

This is a well-written paper in which the authors build upon their previous work by identifying Integrin α11 as a Clec11a/Osteolectin receptor. Further, the authors provide evidence that the anabolic effect of Osteolectin occurs via activation of the Wnt pathway in osteogenic cells. The experiments in this paper are generally well-described, well-documented, and convincingly presented. This paper is appropriate for publication in *eLife* as it significantly advances understanding with regard to factors that regulate adult bone homeostasis.

*Reviewer #3:*

Shen et al. demonstrated that osteolectin contributes to osteogenesis via integrin α11 using biochemical and cellular assays as well as genetical deletion of the integrin in mouse model. It is new to identify integrin α11β1 as an osteolectin receptor and the genetical deletion of integrin α11 phenocopies the osteolectin knockout.

Main discovery is that integrin α11β1 is the osteolectin receptor, which was shown by protein-protein interaction assay. The assay was performed well but the competition assay targeting the interaction of integrin-ligand will intensify the claim. The author developed the hypothesis based on the conserved peptide such as RGD and LDT. There are several inhibitors based on the RGD peptide. Using the inhibitors, the author also can test whether the interaction between two proteins is crucial for osteogenesis in a cell culture experiment. The inhibitors can be useful in vivo experiment as well. The author mainly shows the phenotype of integrin α11 knockout or correlation of β-catenin protein level instead of the interaction between osteolectin and the integrin. Considering the main theme of the paper, I recommend in vivo phenotype test using the ligand and its inhibitor to confirm the functional roles of the interaction between osteolectin and integrin α11β1.

The important feature of Integrin-ligand binding is the phosphorylation of focal adhesion kinase (FAK) at Y397. The phosphorylation initiates integrin-mediated canonical signaling upon ligand binding. However, the author didn't show the level or any description to explain how the interaction initiates downstream signaling. The author used collagen as a control, which is the ligand but not for the osteogenesis/stabilization of β-catenin. Using the level of phosphorylated FAK, the author can show that collagen is active for integrin-ligand binding but the interaction doesn't contribute to osteogenesis.

Another main point is that osteolectin-integrin αV system activates β-catenin signaling to increase osteogenesis. The author showed that 24 hour-treatment of osteolectin stabilizes β-catenin in cells while phosphorylation of GSK3 happened within an hour. The function of β-catenin is well known to be a transcriptional activator but it is also important to maintain cell-cell junction with E-cadherin. The first β-catenin resides in the cytoplasm and the inhibitor of proteasomal degradation of the protein can increase the protein level, which allows the protein to enter the nucleus to activate transcriptional factors. However, the second β-catenin is in the cell membrane and it is not clear whether the level is regulated by the same proteasomal degradation pathway. Therefore, it is necessary to distinguish the levels of β-catenin in cytosol/nucleus from the levels in the membrane by biochemical separation or immunohistochemistry, which will clarify the result. Since author used 24 hours for detection of enhanced β-catenin protein level which is abnormally long compared to Wnt (1-2 hours), it is also necessary to prove that it is from proteasomal degradation not from transcriptional effects or et al. I'd like to suggest pulse-chase of the protein. Additionally, I suggest other time points such as 2 and 4 hours after osteolectin treatment as Wnt treatment experiments.

---

## [Author Response]

Overall there was consensus that the manuscript could break new ground in our understanding of adult stem cell fate and the mechanism of osteolectin in directing lineage through the Itga11 receptor. In general the experiments are illustrative and the results are relatively supportive of the conclusions. However there were several issues in your revision that need to be resolved before a final decision can be rendered. A major concern is the relatively mild skeletal phenotype with the LepRCre Itga11 at 12 months, the lack of studies with inhibitors to define the specificity of the integrin receptor, absence of any integrin signaling studies, concerns about confounding by collagen interactions and details about the recombinant osteolectin experiments in which there were inconsistent Wnt signaling. These issues can certainly be addressed in a revision with a greater 'n' for the in vitro studies, more phenotyping of the conditional model, and experiments related to the role of collagen in modulating integrin signaling, as well as competitive inhibitors for integrin signaling. A summary of the three reviewers comments are noted below:

We thank the reviewers for their thoughtful comments. We provide detailed responses to specific reviewers below, but summarize our responses to the major issues here:

a) “relatively mild skeletal phenotype with the *LepRCre Itga11* at 12 months”

We address this issue in detail in response to reviewer #1. The LepR^+^ cells do not contribute to bone formation during skeletal development or early postnatal growth. LepR^+^ cells only begin to contribute to skeletal bone during adulthood (Cell Stem Cell 15:154). Therefore, conditional deletion of *Itga11* with *Lepr-cre* would not be expected to yield a phenotype at 2 months of age, when only around 5% of osteoblasts derive from LepR^+^ cells (Cell Stem Cell 15:154). The phenotype would not be expected to be apparent until 6-12 months of age, when the LepR^+^ cells are a major source of new osteoblasts (Cell Stem Cell 15:154). We have added new data from four additional pairs of *Lepr-cre; Itga11*^fl/fl^ and sex-matched littermate controls at 12 months of age. The data show that trabecular bone volume and mineral apposition rate were significantly reduced in male and female *Lepr-cre; Itga11*^fl/fl^ mice relative to controls at 6 and 12 months of age. Cortical bone mineral density and mineral apposition rate were significantly reduced in male and female *Lepr-cre; Itga11*^fl/fl^ mice relative to controls at 12 months of age. Therefore, both trabecular and cortical bone were significantly reduced in *Lepr-cre; Itga11*^fl/fl^ mice. The later onset of the cortical bone phenotype as compared to the trabecular bone phenotype was also observed for the *Osteolectin* germline knockout mouse (*eLife* 5:e18782). Therefore, the *Lepr-cre; Itga11*^fl/fl^ mice phenocopy not only the bone loss observed in *Osteolectin* deficient mice but also the difference in timing of trabecular versus cortical bone loss.

b) “lack of studies with inhibitors to define the specificity of the integrin receptor”

We address this below in response to reviewers #1 and #3. We added new data showing that an inhibitor of integrin binding, RGDS peptide, blocked Osteolectin binding to integrin α11β1 (new Figure 1L) and the ability of Osteolectin to induce osteogenic differentiation (new Figure 1M).

c) “absence of any integrin signaling studies”

We address this below in response to reviewer #3. We added new data showing that Osteolectin activates integrin signaling by promoting the phosphorylation of Focal Adhesion Kinase (FAK) at Y397 (new Figure 4G and 4H), in addition to promoting the phosphorylation of GSK3 at Ser21/9 (Figure 2D and Figure 4G and 4H).

d) “concerns about confounding by collagen interactions”

We address this in detail in response to reviewer #1. We did not observe any effect of exogenous collagen on osteogenic differentiation (Figure 1K), and deletion of integrin a11 would be unlikely to block the effects of collagen on LepR^+^ cells since LepR^+^ cells express multiple other collagen receptors.

e) “details about the recombinant osteolectin experiments in which there were inconsistent Wnt signaling”

We address this below in response to Reviewer #1, point #1. We added additional experiments testing the effects of Osteolectin treatment on the transcription of Wnt target genes. The effects are now more consistently statistically significant across experiments.

Reviewer #1:

The authors continue their previous studies on Lepr+ stromal cells and the role of osteolectin in bone formation. Here they explored the receptor for osteolectin and provided evidence that integrin α11 is an osteolectin receptor mediating bone formation. A main concern has to do with the generally mild bone effect of Itga11 deletion by a constitutively active Cre (Lepr-Cre) that acts in not only osteoprogenitors but also many other cell types including adipocytes.

LepR^+^ cells do not contribute to bone formation during skeletal development or early postnatal growth. LepR^+^ cells arise postnatally and do not begin to contribute to bone formation until adulthood (Cell Stem Cell 15:154). Therefore, conditional deletion of *Itga11* with *Lepr-cre* would not be expected to yield a phenotype at 2 months of age, when only around 5% of osteoblasts derive from LepR^+^ cells (Cell Stem Cell 15:154). The phenotype would be expected to become detectable by 6 months of age, when the LepR^+^ cells are a major source of new osteoblasts for skeletal maintenance. The defects observed in the *Lepr-cre; Itga11*^fl/fl^ mice phenocopy the defects observed in *Osteolectin* deficient mice (*eLife* 5:e18782):

We have characterized the *Lepr-cre* recombination pattern in detail (Cell Stem Cell 15:154) and do not observe recombination in other cell types in the bone marrow, where Osteolectin is expressed. *Lepr-cre* recombines in only 0.3% of bone marrow cells. These cells do give rise to adipocytes, but adipocytes are rare in the femurs of young adult mice (the bones we studied) (Cell Stem Cell 15:154). Moreover, adipocytes do not express *Itga11*, as shown in the RNA-seq data in Author response table 1. In light of our in vitro data demonstrating that *Itga11* deficiency acts within LepR^+^ stromal cells to block the osteogenic response to Osteolectin (Figure 7F and 7G), there is no reason to think that the in vivo effects of Osteolectin are instead mediated through an indirect mechanism involving adipocytes.

**Author response table 1. resptable1:** RNA-seq analysis of LepR^+^ cells from non-irradiated bone marrow as well as hematopoietic cells and unfractionated cells from irradiated bone marrow (which are high in adipocytes). *Itga11* is poorly expressed in adipocyte-rich bone marrow cells after irradiation (all data represent average FPKM values from multiple replicates).

Cell type	*Adipoq*	*Fabp4*	*Plin1*	*Itga11*	*Osteolectin*
LepR+ cells from bone marrow	835	23	1.2	34	37
Hematopoietic cells from irradiated bone marrow	1.8	13	0.3	0.1	0.2
Whole bone marrow cells after irradiation (high in adipocytes)	92	1016	68	2.2	2.8

The fact that the bone phenotype does not manifest until much later in life (12 months) makes it very difficult to discern any direct effect. This is an important point as it has become clear that bone mass is regulated by many other tissues outside the skeleton. Furthermore, one cannot discount the potential effect of the integrin deletion on collagen I binding in vivo which may in turn be responsible for the bone phenotype. Other comments are as follows.

The bone phenotype manifests before 6 months of age. We observed a significant reduction in trabecular bone volume (Figure 5G) and trabecular mineral apposition rate (Figure 5M) in the *Lepr-cre; Itga11*^fl/fl^ mice at 6 months of age. Since we would not expect to observe a phenotype until after 2 months of age (see above), and 6 months was the next time point we analyzed, we believe the phenotype became evident in young adult mice, just as would be expected based on the *Osteolectin* deficiency phenotype (*eLife* 5:e18782).

Although many tissues outside the skeleton can influence bone mass, the expression patterns of *Lepr-cre* and *Itga11* are each very restricted. We believe that only osteogenic progenitors express both genes. Moreover, the in vitro data demonstrate a direct effect of Osteolectin on the osteogenic differentiation of LepR^+^ bone marrow stromal cells in culture (Figure 7F and 7G). This effect was completely eliminated when *Itga11* was deleted from LepR^+^ bone marrow stromal cells (Figure 7F and 7G) or from a pre-osteoblast cell line or human bone marrow stromal cells (Figure 4A and 4B). The only conclusion that is consistent with all of the in vitro and in vivo data is that integrin a11 is required in mesenchymal progenitors for the osteogenic response to Osteolectin.

We were unable to detect an effect of exogenous collagen on osteogenesis by three preosteoblast or bone marrow stromal cell lines (Figure 1K). Collagen binding is mediated by several integrins, including a1b1, a2b1, a10b1, and aVb3 (Prog Histochem Cytochem 37:3 and Biochem Biophys Res Commun 182:1025). Since multiple collagen receptors, including a1b1, a10b1 and aVb3, are expressed by bone marrow stromal cells (Figure 1D and 1E), the loss of one integrin (a11) would likely not be sufficient to block the response of these cells to collagen. Even if a11 deficiency had some effect on collagen binding, this would not explain the loss of Osteolectin responsiveness by isolated mesenchymal cells in culture.

"To refine potential candidate receptors for Osteolectin we more closelyexamined the expression of these genes in LepR+ cells, which are osteogenic and should express the receptor, and endothelial cells, which are not osteogenic and should not express the receptor. Itga1, Itga6, and ItgaV were strongly expressed by both LepR+ cells and endothelial cells, suggesting they did not encode the receptor (Figure 1F). Itga11 was expressed exclusively by LepR+ cells, not by endothelial cells or other bone marrow cells (Figure 1F)." The reasoning is not sound as there is no evidence that osteolectin/receptor signaling alone is sufficient to induce osteogenesis. Therefore being expressed in nonosteogenic cells should not be an exclusion criterion for the potential receptor.

We have revised the quoted text to soften the assumption that the Osteolectin receptor would not be expressed by endothelial cells. The expression data were only a way of identifying candidate integrins that might mediate the effect of Osteolectin on osteogenic progenitors. All of our conclusions were based on functional assays that demonstrated that Osteolectin binds α11β1 (Figure 1I and 1J), that a11 is required for the osteogenic response of LepR^+^ cells to Osteolectin in culture (Figure 7F and 7G, Figure 4A and 4B), and that conditional deletion of *Itga11* in LepR^+^ cells in vivo phenocopies the effects of *Osteolectin* deficiency on adult skeletal bone mass (Figure 5F-M and Figure 6A-G). We were unable to detect Osteolectin binding to integrins other than α11β1 and α10β1 (Figure 1I). Nonetheless, we agree that there could potentially be additional Osteolectin receptors that we did not observe, and noted that in the discussion of the manuscript.

Details about production and application of recombinant osteolectin and pro-Collagen I used in Figure 1K are missing.

We have added more detail to the Materials and methods to address this. We purchased recombinant His-tagged human Pro-Collagen 1α from R&D Systems and removed the His tag using TEV protease. After cleavage, we purified the untagged Pro-Collagen 1α using Ni-NTA agarose columns to separate it from the cleaved His tag, the His-tagged Pro-Collagen 1α, and the His-tagged TEV protease.

Was collagen I used to coat the culture plates, which may be necessary to engage the integrins? Some testing of integrin activation is necessary as a positive control in order for any conclusions to be drawn from a negative result here. Same applies to Figure 4D and 4E.

In the data shown in the manuscript, Pro-Collagen 1α or Osteolectin was added to the supernatant of the culture medium. We have added new data to the revised manuscript showing that when Pro-Collagen 1α or Osteolectin was added to the culture medium, each activated integrin signaling in mouse bone marrow stromal cells based on increased FAK and GSK3 phosphorylation (Figure 4G and 4H). However, Osteolectin promoted the accumulation of nuclear β-catenin (Figure 4H and 4I) and osteogenic differentiation (Figure 1K) while ProCollagen 1α did not (Figure 4H, 4I, and 1K). These data suggest that when added to the culture medium, both Pro-Collagen 1α and Osteolectin activate integrin signaling but that Osteolectin is more effective at promoting β-catenin accumulation, Wnt target gene expression, and osteogenic differentiation.

We also tried coating cell culture dishes with Pro-Collagen 1α or Osteolectin before seeding the osteogenic cells. However, when we coated the dishes, neither Pro-Collagen 1α nor Osteolectin promoted osteogenic differentiation.

Also in Materials and methods, details about in vivo application of recombinant osteolectin (dose, age of mice etc.) is missing.

We have added more detail to the Materials and methods to address this (see the ‘Recombinant protein production and application’ section). For in vivo use of recombinant mouse Osteolectin, 50 µg/kg body mass of recombinant Osteolectin was subcutaneously injected daily into 2month-old female *Lepr-cre; Itga11^fl/fl^* or littermate control mice for 28 days.

Figure 2B, Alizarin red staining alone is not sufficient to support claims about OB differentiation especially since the authors deleted an extracellular matrix protein that may directly affect nucleation of minerals. Same criticism applies to Figure 4A, 7C, 7F.

We have added qRT-PCR data to the manuscript showing that the increase in Alizarin red staining in those experiments was also associated with increased transcription of *Dmp1* (see new Figures 2C, 4B, and 7G), a marker of bone formation (JBMR 17-1822).

Figure 2B, the authors observed no increase in β-cat within 1 hr of osteolectin treatment but only after 24 hrs. This data argues against a direct effect of osteolection on β-catenin signaling. Similarly, the role of β-catenin in osteolectin-induced osteogenesis is also likely indirect. Considering the known function of β-catenin in osteoblast differentiation it is expected that β-catenin depletion would suppress osteogenesis by any upstream osteogenic signals.

The time required to detect nuclear β-catenin accumulation after Wnt pathway activation varies among cells and assay conditions. Although some cell types do it within an hour, other cell types (or assay conditions) require more time (J Bone Miner Res 31:2215, Clin Cancer Res 18:4997, Arterioscler Thromb Vasc Biol 34:2268). In the original manuscript, we performed western blots at only 1 and 24 hours, so we knew there was an increase in nuclear β-catenin by 24 hours after Osteolectin treatment but had not done intermediate time points to assess when this was first evident. We have now added new data to the manuscript demonstrating that the increase in nuclear β-catenin was evident within 4-6 hours of Osteolectin treatment (new Figure 4I).

Published studies suggest that β-catenin directly promotes the expression of Wnt pathway target genes that are required for osteogenic differentiation including *Alkaline phosphatase, Axin2, Lef1*, and *Runx2* (Cell 107:513, Mol Cell Biol 22:1172, Nature 382:638, and J. Biol. Chem. 280:33132,). We observed significantly increased transcription of each of these genes, in an a11-dependent manner, after treatment of mesenchymal stem/progenitor cells with Osteolectin in culture (Figure 4D and 4F) and in vivo (Figure 7—figure supplement 1B). The fact that we observe these effects in culture in multiple osteogenic cell lines, as well as primary mouse bone marrow stromal cells, and that the effects of Osteolectin on Wnt target gene expression are abolished by *Itga11* deficiency, strongly argues that Osteolectin/α11β1 signaling directly promotes Wnt pathway activation.

We agree that β-catenin depletion would be expected to block the effects of other osteogenic signals. However, this does not undermine any of our conclusions. The known role of β-catenin in osteogenesis is consistent with our conclusion that Osteolectin promotes osteogenesis at least partly by activating β-catenin. Our data showing that Osteolectin promotes nuclear accumulation of β-catenin (new Figure 4I) and that treatment with IWR-1-endo (a drug that promotes β-catenin degradation) blocks the effect of Osteolectin on osteogenesis (Figure 3C and 3D) suggest that Osteolectin promotes osteogenesis at least partly by activating the Wnt pathway. Other signaling mechanisms may also contribute to osteogenesis in response to Osteolectin but our data show that the increase in β-catenin is necessary. The literature indicates that multiple mechanisms contribute to Wnt pathway activation in osteogenic progenitors (J. Clin. Invest. 116:1202), so there is no reason to think that Osteolectin could not be among them. We have added text to the Discussion more clearly explaining this.

Figure 6. The authors show no change in cortical total area (6B) or cortical area (6C) but a significant decrease in cortical thickness (6D) and Ct Ar/Tt Ar (6E). How do they explain that? Figure 6G, what is the surface used for MAR measurements? In general, how did the authors control for effects of genetic background variations on the parameters?

We have added new data from four additional pairs of *Lepr-cre; Itga11*^fl/fl^ and sex-matched littermate controls at 12 months of age. Now there are no statistically significant differences in cortical total area (Figure 6B), cortical area (Figure 6C), cortical thickness (Figure 6D) or Ct Ar/Tt Ar (Figure 6E). However, *Lepr-cre; Itga11*^fl/fl^ mice do exhibit significantly reduced cortical bone mineral density (Figure 6F) and cortical mineral apposition rate (Figure 6G) at 12 months of age. *Osteolectin* deficient mice also exhibited a milder cortical bone loss as compared to trabecular bone loss, which we only detected in the original manuscript at 16 months of age (*eLife* 5:e18782); therefore, the later onset and milder cortical bone phenotype in *Lepr-cre; Itga11*^fl/fl^ mice phenocopy what we observed in *Osteolectin* deficient mice.

In Figure 6G, the surface used to quantify the cortical mineral apposition rate was the medial endosteal cortical surface of the femur, as previous described (Egan et al., 2012). To avoid effects of genetic background variation, we generated the *Itga11* floxed allele by CRISPR-targeting in pure C57BL/Ka embryos. We confirmed germline transmission of the allele by mating with C57BL/Ka mice, then back-crossed the mice for at least 3 generations to C57BL/Ka wild-type mice prior to analysis of the phenotype. The *Lepr-cre* mice had also been backcrossed for at least 8 generations onto a C57BL/Ka background. Therefore, no mixed genetic backgrounds were used in our experiments. We have revised the Materials and methods to provide this additional detail.

On multiple occasions, the authors seem to imply MC3T3-E1 cells as bone marrow stromal cells, but they are cell lines derived from the newborn mouse calvaria, and generally considered to be more advanced along the OB lineage than the BM stromal cells. These should be corrected. " MC3T3-E1 cells, hBMSC#1 cells, and hBMSC#2 cells secrete Osteolectin into the culture medium (Figure 2A), consistent with our observation that Osteolectin is synthesized by a subset of LepR+ bone marrow stromal cells (Yue et al., 2016)." "Addition of recombinant Osteolectin to culture significantly promoted osteogenesis by parental, but not Itga11 deficient, MC3T3-E1,hBMSC#1, and hBMSC#2 cells (Figure 4A). Integrin α11 is therefore required by mouse and human bone marrow stromal cells to undergo osteogenesis in response to Osteolectin." "Integrin α11 is therefore required by mouse and human bone marrow stromal cells to activate Wnt pathway signaling in response to Osteolectin."

Thank you for catching this error, we have corrected this.

Reviewer #3:

Shen et al. demonstrated that osteolectin contributes to osteogenesis via integrin α11 using biochemical and cellular assays as well as genetical deletion of the integrin in mouse model. It is new to identify integrin α11β1 as an osteolectin receptor and the genetical deletion of integrin α11 phenocopies the osteolectin knockout.Main discovery is that integrin α11β1 is the osteolectin receptor, which was shown by protein-protein interaction assay. The assay was performed well but the competition assay targeting the interaction of integrin-ligand will intensify the claim. The author developed the hypothesis based on the conserved peptide such as RGD and LDT. There are several inhibitors based on the RGD peptide. Using the inhibitors, the author also can test whether the interaction between two proteins is crucial for osteogenesis in a cell culture experiment. The inhibitors can be useful in vivo experiment as well. The author mainly shows the phenotype of integrin α11 knockout or correlation of β-catenin protein level instead of the interaction between osteolectin and the integrin. Considering the main theme of the paper, I recommend in vivo phenotype test using the ligand and its inhibitor to confirm the functional roles of the interaction between osteolectin and integrin α11β1.

Good idea. We have added new data demonstrating that RGDS peptide (Arg-Gly-

Asp-Ser), a potent antagonist of integrin binding (Cell 42:439), inhibited the binding of Osteolectin to integrin α11β1 (new Figure 1L) and blocked the osteogenic response of MC3T3E1 cells and bone marrow stromal cells to Osteolectin in culture (new Figure 1M). We performed these experiments in culture, rather than in vivo, because it would be necessary to inhibit the interaction of Osteolectin with its receptor in vivo for at least a month to detect an effect on osteogenesis. RGDS peptide binds to many other integrins and disrupts many biological processes in vivo, such as the function of integrin αIIbβ3 on platelets and integrins on leukocytes and endothelial cells (Thromb J 15:22, Blood 113:4078 and Cancer Res 62:5139). This would make it challenging to dose the mice with high enough levels of RGDS over a long enough time to detect effects on osteogenesis without causing other serve phenotypes (such as changes in feeding or body mass) that might confound our ability to interpret the results.

The important feature of Integrin-ligand binding is the phosphorylation of focal adhesion kinase (FAK) at Y397. The phosphorylation initiates integrin-mediated canonical signaling upon ligand binding. However, the author didn't show the level or any description to explain how the interaction initiates downstream signaling. The author used collagen as a control, which is the ligand but not for the osteogenesis/stabilization of β-catenin. Using the level of phosphorylated FAK, the author can show that collagen is active for integrin-ligand binding but the interaction doesn't contribute to osteogenesis.

Another good idea, thank you. We have now added new data to the manuscript showing that Osteolectin treatment of mouse bone marrow stromal cells promotes FAK phosphorylation at Y397 to a similar extent as Pro-Collagen 1α (new Figure 4G and 4H). This demonstrates that, although both Osteolectin and Pro-Collagen 1α bound to integrin α11β1 (Figure 1J) and initiated integrin signaling (new Figure 4G and 4H), only Osteolectin promoted nuclear β-catenin accumulation and osteogenic differentiation (new Figure 4H and 4I).

Another main point is that osteolectin-integrin αV system activates β-catenin signaling to increase osteogenesis. The author showed that 24 hour-treatment of osteolectin stabilizes β-catenin in cells while phosphorylation of GSK3 happened within an hour. The function of β-catenin is well known to be a transcriptional activator but it is also important to maintain cell-cell junction with E-cadherin. The first β-catenin resides in the cytoplasm and the inhibitor of proteasomal degradation of the protein can increase the protein level, which allows the protein to enter the nucleus to activate transcriptional factors. However, the second β-catenin is in the cell membrane and it is not clear whether the level is regulated by the same proteasomal degradation pathway. Therefore, it is necessary to distinguish the levels of β-catenin in cytosol/nucleus from the levels in the membrane by biochemical separation or immunohistochemistry, which will clarify the result. Since author used 24 hours for detection of enhanced β-catenin protein level which is abnormally long compared to Wnt (1-2 hours), it is also necessary to prove that it is from proteasomal degradation not from transcriptional effects or et al. I'd like to suggest pulse-chase of the protein. Additionally, I suggest other time points such as 2 and 4 hours after osteolectin treatment as Wnt treatment experiments.

We have added new data to the manuscript in which we treated mouse bone marrow stromal cells with PBS or recombinant Osteolectin then assessed β-catenin levels in the nuclear fraction or the cytosolic/membrane fraction 2, 4, 6, 12, and 24 hours later (new Figure 4I). Osteolectin treatment had no effect on β-catenin levels in the cytosolic/membrane fraction at any time point. However, Osteolectin did significantly increase β-catenin levels in the nuclear fraction after 4, 6, 12, and 24 hours. We also assessed *Ctnnb1* (which encodes β-catenin) transcript levels in the same samples by qRT-PCR and observed no effect of Osteolectin on *Ctnnb1* transcription 6 or 24 hours after Osteolectin treatment (Figure 4J). These data suggest that Osteolectin promotes nuclear β-catenin accumulation within 4 hours in bone marrow stromal cells by inhibiting proteasomal degradation, consistent with Wnt pathway activation.